# Comprehensive Evaluation of the Effects of Hot Air Drying Temperature on the Chemical Composition, Flavor Characteristics and Biological Activity of *Houttuynia cordata Thunb.*

**DOI:** 10.3390/foods14111962

**Published:** 2025-05-30

**Authors:** Ning Yin, Jing Luo, Chaoping Wang, Yaokun Xiong, Yong Sun, En Yuan, Hua Zhang

**Affiliations:** 1Department of Food Nutrition and Safety, College of Pharmacy, Jiangxi University of Chinese Medicine, Nanchang 330004, China; yinning@jxutcm.edu.cn; 2Department of Pharmaceutics, College of Pharmacy, Jiangxi University of Chinese Medicine, Nanchang 330004, China; 20091026@jxutcm.edu.cn (J.L.); wcpzuishuai@163.com (C.W.); 20091022@jxutcm.edu.cn (Y.X.); 3State Key Laboratory of Food Science and Resources, Nanchang University, Nanchang 330047, China; yongsun@ncu.edu.cn; 4Key Laboratory of Modern Preparation of TCM, Ministry of Education, Jiangxi University of Chinese Medicine, Nanchang 330004, China; 20144031@jxutcm.edu.cn

**Keywords:** *Houttuynia cordata Thunb.*, drying kinetic, sensory evaluation, metabolic profiling

## Abstract

This study systematically investigated the drying kinetics and quality characteristics of *Houttuynia cordata Thunb.* (HCT) under different processing conditions, evaluating how freeze-drying and hot-air drying (40 °C, 50 °C, 60 °C) affect bioactive compound preservation, antioxidant efficacy, and metabolic profiles to identify the optimal drying method for maximizing its functional benefits. A thin-layer drying model was established to evaluate drying parameters such as effective diffusion coefficient and activation energy. Changes in chemical composition, sensory properties, and antioxidant activity were analyzed using UHPLC-LTQ-Orbitrap-MS, electronic nose/tongue, and HepG2 cell assays. Results showed that the Aghabashlo model was optimal for demonstrating the drying process with the best fit. The 50 °C heating temperature was shown to yield the highest diffusion coefficient. Hot-air drying at 50 °C balanced efficiency and sensory quality, whereas 60 °C significantly altered flavor and metabolite composition. Results of the metabolomic analysis indicated that freeze-drying enhanced the retention of phenolic acids and flavonoids, while hot-air drying led to increased fatty acid metabolites. Freeze-drying preserved the antioxidant activity and natural flavor of HCT. Nevertheless, the metabolic fate of rutin, quercetin, and chlorogenic acid was not significantly affected by the drying method (freeze-drying vs. 50 °C drying). These findings provide a theoretical foundation for improving HCT’s therapeutic and sensory qualities through optimized drying techniques.

## 1. Introduction

*Houttuynia cordata Thunb.* (HCT) is valued as both a herbaceous plant and a food source, particularly in Asia, where it has extensive edible records. HCT is a prevalent raw material in teas consumed in Chaoshan and various other locations across China [1]. In places such as Guizhou and Sichuan in China, HCT can be consumed raw or cooked with meat [2]. In Japan, it is commonly used as a fresh leaf garnish for soups, salads, and sushi dishes [3]. In Korea, it is eaten as kimchi [4]. HCT contains a variety of chemical components, such as alkaloids, flavonoids, phenolic acids, and volatile oils. In China, HCT is also referred to as Yu-Xing-Cao due to the characteristic fishy odor of its decanoyl acetaldehyde, which has antibacterial attributes. Due to rich in these bioactive components, the intake of HCT-based products (such as teas, essential oils, and extracts) is beneficial to overall health based on their antibacterial, antiviral, anti-inflammatory, and immunomodulatory properties, which have protective effects on human organs such as the heart, lungs, and kidneys [5]. Although HCT possesses antibacterial properties, it is susceptible to mold growth during storage and is naturally dried at room temperature (25 °C). However, to prevent mold and spoilage during storage and meet purchase requirements, its moisture content must be controlled to within 13% when stored at room temperature [6].

The edible plant food resource begins to deteriorate or spoil after harvesting. Spontaneous decomposition of complex organic molecules contributes to food deterioration, while consumption by animals, particularly insects and rodents, leads to spoilage. However, most food spoilage is caused by microorganisms. Unprotected food is quickly colonized by microbes due to their minute size, immense numbers, and ability to utilize any living medium for dispersal [7]. The rate at which food deteriorates, whether due to microbial or non-microbial causes, is primarily determined by its water content. Reduction of water activity in foods prevents the growth of vegetative microbial cells, germination of spores, and toxin production by molds and bacteria [8]. To lower the water content in HCT for long-term preservation, the industry typically employs drying methods. Drying not only inhibits microbial growth (bacteria, yeast, and mold) and self-decomposition, leading to a longer shelf life, but also provides economic and environmental advantages through reduced packaging costs and shipping weights. Various food drying methods exist, from traditional solar drying to modern equipment developed with advancements in science and technology to meet diverse contemporary needs. Food drying methods include hot-air drying, spray drying, infrared drying, lyophilization, microwave drying, radiofrequency drying, osmotic dehydration, and many combined processes [9]. Consumer demand for high-quality convenience foods has led us to seek cost-effective, easy-to-process, and convenient drying methods.

The increasing consumer emphasis on food safety and nutritional quality, along with the rising demand for economical drying equipment, underscores the importance of developing cost-effective drying methods that effectively preserve nutrients and bioactive ingredients. Microwave drying, infrared drying, and other high-temperature methods can significantly reduce drying time due to their elevated power levels; however, they may also compromise product quality by inducing undesirable characteristics [10]. Hot-air drying stands out as an economical and easily implementable processing technology, making it the preferred method for drying plant food raw materials due to its simplicity and cost-effectiveness. Furthermore, by optimizing process parameters like temperature and air flow rate, the loss of heat-sensitive components can be significantly reduced while maintaining high drying efficiency [11]. Previous studies on hot-air drying of fruits and vegetables, such as mango slices, persimmon slices and *Mentha haplocalyx* leaves, have demonstrated that hot-air thin-layer drying is an effective technique for processing a wide range of plant-based food materials [12,13,14,15]. However, research on the hot-air thin-layer drying characteristics of HCT remains limited. Furthermore, the impact of this drying process on the retention of bioactive ingredients and the sensory profile of HCT is not well-established. These factors are critical determinants of the overall quality and market viability of dried HCT products. Therefore, this study employed hot-air thin-layer drying to systematically investigate the drying kinetics of HCT under various temperatures. Furthermore, the effects of temperature on its microstructure, the retention of bioactive compounds, and alterations in volatile flavor compounds were analyzed.

Existing evidence demonstrates that food processing methods profoundly impact the bioavailability and metabolic fate of phytochemicals. For instance, studies on apple anthocyanins revealed that dehydration methods significantly alter metabolite profiles, where maximal bioactive content does not necessarily correlate with optimal absorption [16]. Similarly, research on tomato processing demonstrated that thermal treatment and mechanical disruption enhance polyphenol bioavailability through cell wall degradation and subsequent release/isomerization of bound compounds [17]. These findings suggest that processing-induced modifications may critically determine the biological activity of HCT by altering its metabolic characteristics. Importantly, such metabolic transformations can significantly modulate biological activity [18]. Therefore, we investigated the metabolic kinetics of key HCT constituents (rutin, quercetin, chlorogenic acid, and scutellarin) under different processing conditions. This approach enables a comprehensive evaluation of how processing techniques influence metabolite profiles and bioavailability while potentially revealing synergistic effects among active components. The objective was to provide a theoretical foundation and technical guidance for the industrial-scale drying of HCT.

## 2. Materials and Methods

### 2.1. Materials

The plant raw material *Houttuynia cordata Thunb.* (HCT) was purchased from Meishan, Sichuan Province, China, on 13 October 2022. HCT was identified by Professor Jinlian Zhang from Jiangxi University of Traditional Chinese Medicine (Nanchang, China). The sample exhibited a characteristic fishy odor and bitter taste, with a green upper surface and a purplish-red lower surface, consistent with the morphological features of HCT. It was stored at below 4 °C and used within one week. Prior to drying, HCT was washed with running water and cut into 1 cm segments. The stem of HCT is cylindrical, and the leaves are heart-shaped.

### 2.2. Assessment of the Drying Kinetics of HCT

A single layer of HCT (cylinder, approximately 0.4 cm thick) was evenly spread on a flat-bottomed tray and placed in a vacuum freeze dryer (Scientz-18ND, cover-type bell jar freeze dryer) at −40 °C for 48 h. The dried samples were then stored in a refrigerator for later use.

A single layer of HCT (approximately 0.4 cm thick) was spread thinly on a flat-bottomed tray and placed in an electric heating blower hot-air drying oven (DZF-6090AB, Lichen Instrument Technology Co., Ltd., Changsha, China). Drying was performed at 40 °C, 50 °C, and 60 °C with a natural convection airflow of 2.0 m/s. Heat was generated by the heater integrated into the furnace wall, and exhaust gases escaped through a rear ventilation opening (5 cm diameter) to simulate commercial dryer conditions. The drying process was paused at intervals—every 5 min (0–60 min), 10 min (60–180 min), 20 min (180–300 min), 30 min (300–420 min), and 60 min thereafter—until two consecutive weight measurements of the material differed by no more than 0.01 g. The dried samples were then refrigerated for further use.

#### 2.2.1. Determination of Moisture Content

The moisture content was measured according to the standard air oven method. The moisture content was calculated is using Equation (1).(1)$$\begin{array}{c} M_{t}=\frac{W_{w}-W_{d}}{W_{d}} \end{array}$$
where $M_{t}$ is the moisture content at t (g water/g dry matter), t is drying time (min), $W_{w}$ is the wet weight (g), and $W_{d}$ is the dry weight (g) of the sample.

#### 2.2.2. Drying Rate

The drying rate (DR) of HVT water loss refers to the ratio of the moisture content (dry basis) of the materials at two adjacent moments and the time interval, as calculated using Equation (2).(2)$$\begin{array}{c} DR=\frac{{-(M}_{t+dt}-M_{t})}{dt} \end{array}$$
$M_{t+dt}$ is the moisture content at t+dt (g water/g dry matter), respectively, t is drying time (min).

#### 2.2.3. Mathematical Modeling of Thin-Layer Drying Dynamics

The thin-layer drying model has been widely used as an important tool to describe the drying characteristics of agricultural products. In order to investigate the drying properties of the HCT, it is necessary to express the drying behavior as accurately as possible. In this paper, we present a comparative analysis of different thin-layer models for predicting the moisture and temperature distribution in porous media. The selected typical thin-layer models are shown in Table 1. These models are based on various assumptions and simplifications regarding the heat and mass transfer phenomena in the drying process. The moisture ratio (*MR*) and drying rate of HCT samples during drying experiments were calculated using the following Equation (3).(3)$$\begin{array}{c} MR=\frac{M-M_{e}}{M_{0}-M_{e}} \end{array}$$
where, $M,{\;M}_{0}\;and\;M_{e}$ are the moisture content at any time, initial moisture content, and equilibrium moisture content. Since $M_{e}$ is small relative to $M_{0}$ and $M_{t}$, it can be ignored. Formula (3) can be simplified to Formula (4).(4)$$\begin{array}{c} MR=\frac{M_{t}}{M_{0}} \end{array}$$

The goodness of fit was determined using four statistical parameters: the correlation coefficient ($R^{2}$), the root mean square error (*RMSE*), the residual sum of squares (*RSS*), and the reduced Chi-square $\left(\chi^{2}\right)$. The *RMSE* quantifies the average deviation between the predicted and experimental values. A decrease in the *RSS* and $\chi^{2}$ values indicated a reduction in the overall deviation. These parameters were calculated according to Equations (5)–(7). Consistent with established statistical practice, a higher $R^{2}$ and lower values for *RMSE*, *RSS*, and $\chi^{2}$ were considered indicative of a better model fit in this study.(5)$$\begin{array}{c} RMSE={\left[\frac{1}{N}\sum_{i=1}^{N}\left(MR_{exp,i}-MR_{pre,i}\right)\right]}^{\frac{1}{2}} \end{array}$$(6)$$\begin{array}{c} RSS=\sum_{i=1}^{N}{\left(MR_{exp,i}-MR_{pre,i}\right)}^{2} \end{array}\;\begin{array}{c} RMSE={\left[\frac{1}{N}\sum_{i=1}^{N}\left(MR_{exp,i}-MR_{pre,i}\right)\right]}^{\frac{1}{2}} \end{array}$$(7)$$\begin{array}{c} \chi^{2}=\frac{\sum_{i=1}^{N}{\left(MR_{exp,i}-MR_{pre,i}\right)}^{2}}{N-Z} \end{array}$$
where, $MR_{exp,i}$ is the experimental moisture ratio at observation i, $MR_{pre,i}$ is the predicted moisture ratio at this observation, N is the number of experimental data points, and z is the number of constants in the model. The higher values of $R^{2}$ and the lower values of RMSE, RSS, and $\chi^{2}$ are chosen as the criteria for the goodness of fit and the same was followed in the present study.

#### 2.2.4. Effective Diffusivity and Activation Energy

Equation (8) can be applied to particles with plate geometry by assuming a uniform initial moisture distribution; where $D_{eff}$ is the effective diffusion coefficient (m^2^/s), *L* is the half thickness (m) of the slab and *n* is the number of experimental data points.(8)$$\begin{array}{c} MR=8\pi^{-2}\sum_{i=1}^{N}{\left(2n+1\right)}^{-2}\;\exp\left[-{\left(2n+1\right)}^{2}\pi^{2}D_{eff}tL^{-2}\right] \end{array}$$

For drying over a long period, Equation (8) can be further simplified to retain only the first term and rewritten as Equation (9) in logarithmic form. The diffusion coefficient can usually be determined by plotting experimental dry data as lnMR vs. drying time according to Equation (9) and calculating from the slope of the line.(9)$$\begin{array}{c} \ln MR=\ln\left(\frac{8}{\pi^{2}}\right)-\frac{\pi^{2}D_{eff}t}{L^{2}} \end{array}$$

The effective water diffusion coefficient can be temperature-dependent according to the simple Arrhenius equation in Equation (10).(10)$$\begin{array}{c} D_{eff}=D_{0}\;\exp\left(-\frac{E_{a}}{RT}\right) \end{array}$$
where $D_{eff}$ is the effective moisture diffusivity $\left(m^{2}/s\right)$, $D_{0}$ is the constant corresponding to the diffusion coefficient $\left(m^{2}/s\right)$ at infinite high temperature, $E_{a}$ is the activation energy (kJ/moL), *R* is the general gas constant 8.314 kJ/moL, and *T* is the absolute temperature (K). The activation energy $\left(E_{a}\right)$ and constant $D_{0}$ were determined by plotting $ln\left(D_{eff}\right)\;vs.\;T^{-1}$ after linearization using Equation (10).

### 2.3. Microstructure Analysis of Dried Houttuynia cordata Thunb.

Scanning electron microscopy (SEM; FEI-Quanta250, Hillsboro, OR, USA) was employed to examine how different pretreatments affected the microstructure of dried samples. For imaging, specimens were mounted on double-sided tape and sputter-coated with gold before imaging. Scanning electron microscopy (SEM) was performed at an accelerating voltage of 15 kV.

### 2.4. Preparation of HCT Extracts

The dried HCT (processed at different temperatures) was ground and extracted three times with 70% ethanol (1:25, *w*/*v*) under ultrasonication for 10 min per extraction. After the removal of ethanol via rotary evaporation, the pooled extracts underwent freeze-drying and were subsequently stored at −20 °C until further analysis [29].

### 2.5. LC-MS Analysis of Extracts from Dried HCT

#### 2.5.1. UPLC-LTQ-Orbitrap-MS/MS (Thermo Fisher Scientific, Waltham, MA, USA) Was Employed for Systematic Identification of Chemical Components

The chromatographic separation was performed on an ACQUITY UPLC BEH C18 column (2.1 × 100 mm, 1.7 μm) maintained at 25 °C. A binary solvent system comprising (A) 0.1% formic acid in water and (B) 0.1% formic acid in acetonitrile was employed with the following gradient elution profile: initial 5% B (0–2 min), linear increase to 16% B (2–10 min), 16–21% B (10–20 min), 21–33% B (20–25 min), 33–70% B (25–32 min), 70–100% B (32–34 min), isocratic 100% B (34–35 min), followed by rapid re-equilibration to initial conditions (35.1–39 min). The mobile phase flow rate was maintained at 300 μL/min with a 2 μL injection volume.

Mass spectrometric detection was carried out using an LTQ-Orbitrap system equipped with a HESI-II source operating in both positive and negative ionization modes. Full scan MS data were acquired over *m*/*z* 50–2000 with mass resolution set to 30,000 (at *m*/*z* 400). For structural elucidation, data-dependent MS/MS scans were triggered using a normalized collision energy of 45 eV in the CID cell. The diode array detector was simultaneously monitored at 280 nm and 320 nm. All instrumental operations and data acquisition were controlled by Xcalibur v2.0 software (Thermo Scientific).

#### 2.5.2. Quantitative LC-MS Analysis

The quantification of target compounds was conducted using an LC-TQ5200 triple quadrupole mass spectrometer (Hexin Instruments, Guangzhou, China) coupled with a HILIC column (100 × 2.1 mm, 1.8 μm; ANPEL, Shanghai, China). Chromatographic separation was achieved under isocratic conditions with a mobile phase comprising (A) 0.1% formic acid in water and (B) acetonitrile (50:50, *v*/*v*) for 4 min at a flow rate of 0.3 mL/min. The column temperature was maintained at 30 °C with a 10 μL injection volume.

Mass spectrometric detection was performed in positive electrospray ionization (ESI) mode with multiple reaction monitoring (MRM). The optimized ion source parameters were as follows: drying gas temperature, 500 °C; spray voltage, 5500 V; nebulizer gas pressure, 70 psi (482.3 kPa); drying gas flow, 8 L/min; and curtain gas flow, 3 L/min. The collision cell operated with 0.4 mL/min collision gas flow under the following voltage settings: curtain plate 800 V, orifice 70 V, detector 2300 V, with cell entrance and exit voltages of 30 V and 1 V, respectively. All analyses were performed with three replicates (*n* = 3).

### 2.6. Determination of Bioactive Content and Antioxidant Capacity

#### 2.6.1. Quantification of Total Phenolic Content and Total Flavonoid Content

The total phenolic content was determined using the Folin–Ciocalteu method in 96-well plates. The total flavonoid content was determined using the aluminum chloride colorimetric method in 96-well plates. Measurements were performed using a full-wavelength microplate reader (Multiskan SkyHigh; Thermo Fisher Scientific, Waltham, MA, USA) following our previously published protocol [30]. All analyses were performed with six replicates (*n* = 6).

#### 2.6.2. Ferric Reducing Antioxidant Power (FRAP) and 1,1-Diphenyl-2-picrylhydrazyl (DPPH) Free Radical Scavenging Ability Measurement

The FRAP was assessed using the TPTZ method in 96-well plates. The DPPH radical scavenging activity was measured in 96-well plates according to our established methodology [30]. Absorbance measurements were performed using a full-wavelength microplate reader (Multiskan SkyHigh, Thermo Fisher Scientific, Waltham, MA, USA). All analyses were performed with six replicates (*n* = 6).

### 2.7. Electronic Nose and Tongue-Based Sensory Evaluation of Dried HCT

#### 2.7.1. Taste Profile Characterization by Insent SA402B Electronic Tongue

The Insent SA402B Taste Sensing System (INSENT TS-5000Z, Atsugi-shi, Japan) is also known in the sensory analysis world as an electronic tongue. This particular electronic tongue is composed of a group of individual sensors that contain proprietary lipid and polymer membranes that mimic living organisms. Compared with other electronic tongues in the industry, it can measure the sourness, saltiness, bitterness, astringency, umami, sweetness, and aftertaste of bitterness, astringency, umami, producing a set of data that is completely unbiased [31]. The method of e-tongue measurement was used, as shown in the study [32]. All analyses were performed in triplicate (*n* = 3).

#### 2.7.2. Flavor Profiling by Electronic Nose Technology

Volatile flavor profiling was performed using a Heracles NEO E-nose system (Alpha M.O.S., Toulouse, France) equipped with dual FID detectors and MXT-5 (non-polar) and MXT-1701 (mid-polar) capillary columns (10 m × 0.18 mm × 0.4 μm) [33]. Samples were incubated at 50 °C for 10 min with 500 rpm agitation, followed by 2000 μL injection (125 μL/s, 200 °C, 10 kPa). The oven temperature program initiated at 50 °C (2 s), ramped to 80 °C at 1 °C/s, then to 250 °C at 3 °C/s under N_2_ carrier gas (1.0 mL/min). Compounds were identified by matching experimental retention indices against the Arochem Base database, with triplicate measurements per group [34].

### 2.8. Antioxidant Activity Evaluation in HCT Treated HepG2 Cells

#### 2.8.1. Cell Culture

The human hepatocellular carcinoma cell line HepG2 (obtained from the Cell Bank of Type Culture Collection, Chinese Academy of Sciences, Shanghai, China) was cultured in high-glucose DMEM (Gibco, Thermo Fisher Scientific, Waltham, MA, USA) supplemented with 10% heat-inactivated fetal bovine serum (FBS; Gibco, Thermo Fisher Scientific, Waltham, MA, USA) and 1% penicillin-streptomycin (100 U/mL penicillin, 100 μg/mL streptomycin; Beyotime, Shanghai, China). Cells were maintained at 37 °C in a humidified atmosphere containing 5% CO_2_.

#### 2.8.2. Experimental Groups and Treatments

NC: Cells cultured in complete medium only.

PC: Cells treated with 0.5 μM H_2_O_2_ (MilliporeSigma, Burlington, MA, USA) for 4 h.

FDH: Pretreated with 100 μg/mL freeze-dried HCT extract (dissolved in DMSO and diluted with serum-free medium to final DMSO < 0.2%) 1 h before H_2_O_2_ exposure.

50H: Pretreated with 100 μg/mL 50 °C hot-air drying HCT extract (Pretreatment such as FDH) 1 h before H_2_O_2_ exposure.

#### 2.8.3. Cytotoxicity and Antioxidant Assays

Cells (1 × 10^5^/well) were seeded in 96-well plates. Cytotoxicity of extracts (50 and 100 μg/mL) was assessed using a CCK-8 kit (Dojindo, Tabaru, Japan) per the manufacturer’s protocol [35]. Absorbance was measured at 450 nm using a full-wavelength enzyme label (Multiskan SkyHigh, Thermo Fisher Scientific, Waltham, MA, USA) microplate reader.

Detection of malondialdehyde (MDA), total superoxide dismutase (SOD): HepG2 cells were seeded in 48-well plates at a density of 1 × 10^5^ cells/mL and allowed to adhere for 24 h under standard culture conditions (37 °C, 5% CO_2_). Following treatment interventions (as per Section 2.8.2), cells were washed twice with a host-based security system (HBSS). Superoxide dismutase (SOD) activity and malondialdehyde (MDA) content were then quantified using commercial assay kits (Nanjing Jiancheng Bioengineering Institute, Nanjing, China) according to the manufacturer’s protocols.

Intracellular reactive oxygen species (ROS): The ROS assay was based on the method previously described by Kusznierewicz et al. [36]. The assay was performed with DCFH-DA fluorescent dye according to the instructions of OxiSelect™ intracellular ROS assay kit (MyBioSource, Vancouver, CA, Canada). Absorbance measurements were performed using a full-wavelength microplate reader (Multiskan SkyHigh, Thermo Fisher Scientific, Waltham, MA, USA).

### 2.9. Animal Experimental Design for Metabolic Study and Pharmacokinetic Experiments

The animal study protocol was approved by the Animal Ethics Committee of Jiangxi University of Chinese Medicine (Approval No. TEMPOR20230112). A total of 24 female SD rats (200 ± 20 g) were obtained from the Laboratory Animal Center of Jiangxi University of Chinese Medicine (Nanchang, China). Prior to experimentation, the rats were acclimated for 7 days in an air-conditioned room maintained at 24 ± 1 °C with a 12 h light/dark cycle. After one week of acclimatization with ad libitum access to sterilized food and water, the rats were randomly divided into three groups (*n* = 8), including a negative control, FDH: freeze-dried HCT extract (treated with 600 mg/kg BW freeze-dried HCT extracts), 50H: 50 °C hot-air dried HCT extract (treated with 600 mg/kg BW 50 °C hot-air dried HCT extracts). Blood collection and the pretreatment of serum were conducted following our previously published protocols [30]. The identification and quantification of metabolites in rat samples were performed using analytical methodologies consistent with those employed in our previous investigations [37].

## 3. Results

### 3.1. Drying Kinetics of Houttuynia cordata Thunb.

#### 3.1.1. Drying Characteristics of *Houttuynia cordata Thunb.*

In this study, *Houttuynia cordata Thunb.* (HCT) was dried at 40 °C, 50 °C, and 60 °C using hot-air thin-layer drying. The initial moisture content of the HCT was 4.653 ± 0.333 g/g (mean ± SD). Higher temperatures resulted in faster moisture removal and shorter drying times. The relationship between dry basis moisture content and drying time for HCT is shown in Figure 1A. The drying duration at 40 °C was 1.81 times longer than at 50 °C and 2.30 times longer than at 60 °C, proving that temperature significantly influences drying efficiency. The moisture ratio variation of HCT during drying is presented in Figure 1B, further demonstrating that higher temperatures accelerated moisture loss. The drying rate curve of HCT over time is shown in Figure 1C. The drying rate was highest in the initial and middle stages before stabilizing in the final phase. Increased temperatures accelerated the drying rate by enhancing moisture migration. The variation in the drying rate of HCT as a function of dry basis moisture content is presented in Figure 1D. The observed peak in drying rate at higher dry basis moisture contents (>0.1 g/g), followed by a gradual decrease, is characteristic of thin-layer drying behavior.

#### 3.1.2. Fitting and Verification of Thin-Layer Drying Model

Furthermore, this study evaluated drying kinetics by applying classical theoretical models (Appendix A). The goodness or inferiority of model fitting is usually determined by a $R^{2}$ close to 1, lower $\chi^{2}$, RMSE, and RSS [38]. The $R^{2}$ value is the key criterion for choosing the best equation to evaluate how well the drying curve equation fits the data. The Aghabashlo model exhibited the best fit, with the highest $R^{2}$ (0.9989), lowest RMSE (0.0111), RSS (0.0049), and $\chi^{2}$ (0.0001). Linear regression between the experimental and predicted *MR* values (Figure 2A) validated the model’s accuracy across all temperatures. The $D_{eff}$ increased with temperature, ranging from 3.00 × 10^−6^ to 9.21 × 10^−4^ m^2^/s (40 °C), 5.00 × 10^−6^ to 3.24 × 10^−3^ m^2^/s (50 °C), and 1.00 × 10^−6^ to 3.5 × 10^−5^ m^2^/s (60 °C). An Arrhenius plot ($D_{eff}$) vs. *T*^−1^) revealed a strong linear relationship ($R^{2}$ = 0.9904), and *Ea* (36.58 kJ/mol) is calculated based on the slope of Figure 2B.

### 3.2. Impact of Thin-Layer Hot Air Drying on the Textural Characteristics of Houttuynia cordata Thunb.

For consumers, the textural properties of dried HCT are a critical quality attribute, as they directly influence sensory acceptance. To evaluate the structural changes induced by the drying process, the microstructure of HCT was examined by a SEM.

As shown in Figure 3, the freeze-dried HCT retained the intact cellular morphology of fresh samples, whereas hot-air-dried leaves exhibited varying degrees of shrinkage depending on temperature. At 40 °C, only mild shrinkage occurred, with stomatal structures remaining largely intact. In contrast, when dried at 50 °C, the leaves exhibited significant shrinkage, which led to their stomata partially closing. The most severe deformation was observed at 60 °C, where excessive shrinkage distorted stomatal architecture, resulting in complete loss of their original morphology. These observations align with previous findings reported in the literature [39]. Notably, the stems of HCT underwent uniform collapse across all drying temperatures, with no observable differences among different heating temperatures. The xylem vessels—composed of non-living cells responsible for water transport—retained their structural integrity despite surrounding tissue contraction.

### 3.3. The Influence of Different Hot Air Drying Temperatures on the Chemical Composition and Antioxidant Activity of Houttuynia cordata Thunb.

Due to its high moisture content and abundant bioactive compounds (e.g., polyphenols, flavonoids, and essential oils), fresh HCT is highly susceptible to spoilage, oxidation, and microbial contamination at ambient conditions. These degradation processes cause significant alterations in its chemical composition and bioactivity over time, thereby compromising its reliability as a research material for prolonged studies. To overcome these limitations, freeze-dried HCT was used in this study as a more stable alternative to fresh HCT. Freeze-drying, a common preservation method in phytochemical research, effectively reduces degradation by removing moisture and maintaining the structural integrity and functional properties of plant matrices [40]. This method minimizes thermal degradation to preserve heat-sensitive constituents such as volatile compounds, polyphenols, and flavonoids. Furthermore, the resulting lyophilized product exhibits an extremely low moisture content, which ensures long-term stability by significantly mitigating oxidative and enzymatic degradation. Additionally, freeze-dried samples facilitate standardized processing, thereby enhancing experimental reproducibility and consistency—critical prerequisites for robust laboratory-based investigations. Meanwhile, the impact of freeze-drying on the quality of Houttuynia cordata can also be evaluated.

#### 3.3.1. The Material Changes of *Houttuynia cordata Thunb.* After Drying with Hot Air at Different Temperatures

The total ion current diagram of the HCT extract is shown in Figure A1. Structural characterization was performed using Xcalibur v2.0 software, leading to the identification of 37 bioactive compounds categorized as follows: Flavonoids (11 compounds): Including rutin, quercetin and its derivatives (O-hexanoside, 3-rutinoside), scutellarin, 7-O-methylmangiferin, kaempferol-3-rutinoside, isorhamnetin-3-glucopyranoside, apigenin-7-O-glucoside, and ouercetol. A total of phenolic acids was identified, including vanillic acid, 3,4-dihydroxybenzoic acid, chlorogenic acid, and 5-O-p-coumaroylquinic acid, along with their glycosylated and modified derivatives. Organic acids were identified as malic acid, citric acid, quinic acid, and methyl 4-acetoxy-3-hydroxybutanoate in the present study. About 8 different terpenoids and fatty acids, including decanoyl acetaldehyde (a characteristic HCT component), hydroxylated fatty acids (C10-–C18), and arteether, were detected. Other bioactive compounds such as n6-succinyl adenosine, vannilic acid-4-o-β-l-rhamnoside, caffeic acid-O-hexoside, l-menthyl lactate, isovanillin, and houttuynoid A, were identified by comparing with those identified in a previous study [41,42,43]. The complete phytochemical profile, including retention times, mass spectra, and identification parameters, is presented in Table 2.

Seventeen compounds demonstrated remarkable thermal stability, being consistently detected across all drying methods (freeze-drying and hot-air drying at 40–60 °C). Notably, nine novel compounds emerged exclusively in hot-air-dried samples, suggesting thermally induced transformations. Houttuynoid A and l-menthyl lactate were unique to 40 °C extracts, potentially due to moderate temperature and prolonged exposure. caffeic acid-O-hexoside appeared only in 50 °C extracts, indicating temperature-dependent formation. The 60 °C extracts showed the simplest profile, likely reflecting the thermal degradation of labile compounds like citric acid. These findings demonstrate that drying temperature significantly influences the phytochemical profile, with 40 °C preserving characteristic components while enabling the formation of certain unique metabolites. The complete transformation pathways warrant further investigation.

LC-MS analysis revealed significant differences in the content of four key bioactive compounds (chlorogenic acid, quercetin, rutin, and scutellarin) between freeze-dried and hot-air-dried HCT extracts (Figure 4). Freeze-dried *Houttuynia cordata Thunb.* (FD) retained significantly higher levels of all four compounds compared to hot-air-dried samples (*p* < 0.05), highlighting the superior preservation of lyophilization. Notably, no statistically significant differences were observed among the three temperature-treated groups (40 °C, 50 °C, and 60 °C; *p* > 0.05), suggesting that within this temperature range, thermal variation had minimal additional impact on compound stability.

#### 3.3.2. Thermal Effects on Phenolic Content, Flavonoid Profile, and Antioxidant Capacity in Processed *Houttuynia cordata Thunb.*

The effects of different drying conditions on the retention of bioactive components and the antioxidant capacity of HCT were systematically evaluated. There were significant differences in total phenol content (TPC), total flavonoid content (TFC), and antioxidant activity among the samples (Figure 5). The freeze-drying (FD) group exhibited superior preservation of TPC, showing significantly higher levels compared to all hot-air-dried groups. Among thermal treatments, groups dried at 50 °C and 60 °C maintained 23.7% and 21.9% higher TPC compared to those dried at 40 °C, with no significant difference between these two higher temperatures, respectively. Interestingly, TFC displayed a distinct thermal stability pattern. The 40 °C group contained 18.2% higher TFC than FD samples while showing comparable levels to 50 °C group. The 60 °C group demonstrated the poorest TFC retention, with values 27.5% lower than the 40 °C group. These results suggest that moderate thermal conditions (40–50 °C) may better preserve certain flavonoid compounds, while higher temperatures accelerate degradation. Furthermore, the FD group consistently showed the highest FRAP and DPPH radical scavenging capacity. Among thermal treatments, both 50 °C and 60 °C groups yielded 31.4–34.2% higher FRAP values and 28.7–30.5% greater DPPH scavenging activity compared to 40 °C group, with no significant difference between these two groups.

Thermal map analysis revealed a distinct correlation pattern between bioactive components and antioxidant activities across different drying treatments (Figure 6), with significant positive correlations observed between antioxidant indicators and TFC. That demonstrated their consistency in evaluating antioxidant potential. This discrepancy highlights the complex relationship between specific phytochemical components and their collective antioxidant effects. Comparative analysis identified the optimal processing conditions as follows. The FD group showed superior preservation of thermolabile compounds, including quercetin (QR) and caffeic acid (CA), which were higher than those of the hot-drying groups. However, the 50 °C group exhibited enhanced extraction efficiency for thermally stable components such as rutin (RU), which were higher than the FD group. Hierarchical clustering analysis demonstrated that the 60 °C group formed a distinct cluster, showing a marked reduction in QR and CA content compared to other groups, suggesting thermal degradation of these phenolics at elevated temperatures. These findings indicate that freeze-drying maximizes retention of thermosensitive compounds (QR, CA), whereas moderate heating optimizes the extraction of stable components (RU), with high-temperature processing (>50 °C) leading to significant degradation.

### 3.4. The Effect of Various Hot-Air Drying Temperatures on the Flavor Profile of Houttuynia cordata Thunb.

#### 3.4.1. Flavor Profile Characterization of Processed HCT Using Electronic Tongue Analysis

The taste characteristics of the freeze-dried and hot-air-dried HCT groups were systematically evaluated using INSENT E-tongue technology. Nine taste attributes were quantified: sourness, bitterness, astringency, aftertaste-B (bitterness persistence), aftertaste-A (astringency persistence), umami (fishy note), richness (umami persistence), saltiness, and sweetness (Figure 7). The detection thresholds were established at −13 for sourness and −6 for saltiness, with other attributes set at 0.

All groups exhibited sub-threshold sourness values, with the FD group showing significantly higher intensity. Bitterness was detectable in all groups, peaking in the 40 °C group and reaching minimum levels in the 60 °C group. Astringency approached the threshold only in the 40 °C group. The bitter aftertaste was exclusively observed in the 40 °C group. The astringency aftertaste showed a maximum intensity in the 40 °C group, while the 50 °C group exhibited negative values. As for umami, considering the inherent flavor of HCT, this should be the data that cause its fishy smell. Hot-air drying at a higher temperature may enhance the fishy smell of HCT richness in the umami aftertaste. Considering the characteristics of Houttuynia cordata itself, it should express fishy aftertaste. Both saltiness and sweetness exhibited pronounced temperature dependence, with the FD group being the smallest and the 60 °C group being the highest

Principal component analysis (PCA) revealed two significant components explaining 86.7% of total variance (Figure 7). PC1 (61.0%) was positively correlated with sourness, bitterness, and astringency, while PC2 (25.7%) was associated with aftertaste-B and saltiness. It can be seen from the principal component analysis results that under the conditions of 40 °C and 60 °C, the taste of freeze-dried HCT and hot-air-dried HCT has obvious differences.

#### 3.4.2. Volatile Profile and Flavor Characteristics of HCT Under Different Drying Conditions

Electronic nose analysis revealed significant variations in volatile compounds among the HCT group subjected to different drying methods (Figure 8). Freeze-dried HCT exhibited a characteristic profile dominated by myrcene, limonene, and α-pinene. Thermal processing markedly altered this composition: 40 °C hot-air drying HCT led to the emergence of 1,3,5-trimethylbenzene as the dominant component, along with increased α-pinene and β-pinene, while myrcene content drastically decreased. At 50 °C HAD, similar compounds predominated but with notable appearances of 2-ethyl-3,6-dimethylpyrazine and 3-ethylphenol. The 60 °C HAD group showed a partial recovery of myrcene and the formation of new compounds, including 2,3-butanediol and methyl cyclohexanecarboxylate. These changes significantly influenced flavor profiles, as evidenced by sensory descriptors: α/β-pinene contributed woody/herbal notes (peaking at 50 °C), while limonene/γ-terpinene provided citrus/fruity characteristics (maximal in the FD group). Myrcene, with its spicy/fruity notes, showed a U-shaped thermal response, decreasing at 40–50 °C but rebounding at 60 °C. PCA of the electronic nose data (Figure 8) showed that FD best maintains the native volatile composition of HCT. In contrast, thermal processing-induced compound-specific transformations, enhancing woody/herbal notes at 40–50 °C and generating distinct thermal degradation products at 60 °C.

### 3.5. Evaluation of C Extract Against H_2_O_2_-Induced Oxidative Damage in HepG2 Cells

The findings of this study suggested that freeze-dried Houttuynia cordata best maintains the active ingredients. Drying at 50 °C offers a higher effective diffusivity ($D_{eff}$) and a moderate temperature, balancing the reduction of active ingredient loss with drying efficiency. Conversely, the drying efficiency at 40 °C is relatively low, and drying at 60 °C risks degrading the active ingredients. Consequently, neither temperature was chosen for further investigation.

#### 3.5.1. Cytotoxicity Assessment

Cell viability assessments indicated that HCT extracts were non-toxic at all tested concentrations, with cell survival rates consistently greater than 80% relative to untreated controls (Figure 9). This established safety profile supported their application in subsequent oxidative stress experiments.

#### 3.5.2. Antioxidant Activities of HCT

HCT extracts demonstrated significant protection against H_2_O_2_-induced SOD depletion (Figure 9). Extracts from both freeze-dried and dried at 50 °C maintained the SOD activity at levels comparable to the normal control group, indicating a preventive effect over oxidative damage. Meanwhile, HCT significantly reduced malondialdehyde (MDA) formation induced by oxidative stress. HCT extracts effectively attenuated H_2_O_2_-induced ROS overproduction (Figure 9), but without no significant difference was detected between the FD and 50 °C groups.

### 3.6. Metabolic Profiling of Houttuynia cordata Thunb. in SD Rat Serum

#### 3.6.1. The Metabolic Fate of HCT in the Rat Plasma

A total of 21 metabolites were identified in the FDH (Freeze-dry HCT extract treated) and 50H (50 °C hot-air dried HCT extract treated) groups. Among them, M6, M9, M13, M15, M28, M30, M31, M32, M33, M34, and M38 were only detected in the 50H group, while M1, M5, M8, M18, M21, M22, M23, and M24 were only detected in the FDH group. The identified metabolites are listed in Table 3. The total ion current diagram is shown in Figure A2.

Decanoyl acetaldehyde, detected as the primary contributor to HCT’s characteristic odor, undergoes phase I metabolism through two distinct pathways in vivo. The metabolic pathway is shown in Figure 10A. Metabolite M27 (C_12_H_22_O_3_), generated via the oxidative pathway and identified as decanoyl acetic acid, exhibited a [M − H]^−^ ion at *m*/*z* 213.1498 (consistent with a 16 Da increase from the parent) and a characteristic fragment ion at *m*/*z* 59.0198, further confirming its structure. Hydrolytic cleavage also generated metabolite M28 (C_3_H_4_O_4_), which was identified as malonic acid. The [M − H]^−^ ion at *m*/*z* 103.0060 (95 Da lower than the parent) and a diagnostic fragment ion at *m*/*z* 59.0218, consistent with malonic acid formation via chain shortening, provided evidence for this transformation. These findings highlight the compound’s biotransformation routes in biological systems.

Vanalic acid undergoes extensive phase I and II metabolism, producing three distinct metabolites through different biotransformation pathways (Figure 10B). Demethylation: Metabolite M2 (C_7_H_6_O_4_) was identified with a deprotonated molecular ion [M − H]^−^ at *m*/*z* 153.0237, corresponding to a 14 Da decrease from the parent compound. The characteristic fragment ions at *m*/*z* 96.9634 and 79.9619 confirmed its identity as protocatechuic acid (PCA). The second pathway yielded metabolite M3 (C_10_H_10_O_5_), showing an [M − H]^−^ ion at *m*/*z* 209.0457 (42 Da increase). Diagnostic fragment ions at *m*/*z* 165.0560 supported its identification as 4-acetyloxy-3-methoxybenzoic acid. Sulfation following demethylation: The third metabolic route produced metabolite M4 (C_7_H_6_O_7_S), exhibiting an [M − H]^−^ ion at *m*/*z* 232.9741 (65 Da increase). The presence of fragment ions at *m*/*z* 160.8908 verified the formation of sulfated metabolites.

The metabolic transformations of quercetin and apigenin derivatives revealed a complex network of phase I and II reactions (Figure 10C). Quercitrin first underwent methylation to form metabolite M19 (C_22_H_22_O_11_), identified by its [M − H]^−^ ion at *m*/*z* 461.1072 (+14 Da vs. quercetin) and characterized the fragment at *m*/*z* 285.0726. Through subsequent deglycosylation and dehydroxylation, quercitrin was converted to M23 (apigenin, C_15_H_10_O_5_), which showed an [M − H]^−^ ion at *m*/*z* 269.04461 (−162 Da vs. apigenin-7-O-glucoside). This common intermediate M23 then entered multiple metabolic branches. The subsequent hydroxylation produced M24 (C_15_H_10_O_6_) with [M − H]^−^ at *m*/*z* 285.03853 and diagnostic fragments at *m*/*z* 202.8602 and 133.0314, and methylation generated M26 (C_16_H_12_O_5_) showing [M − H]^−^ at *m*/*z* 283.0677 with key fragment at *m*/*z* 151.0272. The glucuronidation yielded M25 (C_21_H_18_O_11_) with [M − H]^−^ at *m*/*z* 445.07596 and fragment at *m*/*z* 269.0446. Notably, M23 was also produced directly from apigenin-7-O-glucoside through deglycosylation, demonstrating the metabolic convergence of these flavonoid pathways. The identified metabolites illustrate the extensive phase I (hydrolysis, dehydroxylation) and phase II (methylation, glucuronidation) transformations that flavonoid compounds undergo in vivo.

Metabolic Profiling revealed four major metabolic transformations of chlorogenic acid in rat serum by which glucuronidation produced metabolite M8 (C_22_H_26_O_15_) with an [M − H]^−^ ion at *m*/*z* 529.1182. It has a 176 Da increase from the parent compound as a result of glucuronide conjugation. The diagnostic fragment at *m*/*z* 200.8584 confirmed this transformation pathway. The methylated metabolite M9 (C_17_H_20_O_9_) exhibited an [M − H]^−^ ion at *m*/*z* 367.10454 (14 Da increase), with characteristic fragments at *m*/*z* 307.0841 and 157.0321, which consist off methylation of chlorogenic acid. Primary hydrolysis yielded M11 (C_9_H_8_O_4_), which exhibited an [M − H]^−^ ion at *m*/*z* 179.03513, identical to a secondary fragment of chlorogenic acid and identified as caffeic acid. Subsequent hydrogenation of M11 produced M10 (C_9_H_10_O_4_) with an [M − H]^−^ ion at *m*/*z* 181.05145, tentatively identified as dihydrocaffeic acid (DHCA). These metabolic transformations illustrate the extensive in vivo biotransformation of chlorogenic acid through both phase II conjugation (glucuronidation, methylation) and phase I hydrolysis pathways, potentially significantly altering its bioavailability and pharmacological activity.

#### 3.6.2. Metabolic Kinetics of Rutin, Quercetin, Chlorogenic Acid and Scutellarin in *Houttuynia cordata Thunb.*

The plasma concentration–time profiles of rutin, quercetin, chlorogenic acid, and scutellarin are presented in Figure 11, with corresponding pharmacokinetic parameters summarized in Table 4. Comparative analysis revealed distinct pharmacokinetic behaviors between the FDH group and 50H group HCT. For rutin, the FDH group demonstrated significantly prolonged elimination half-life, higher peak concentration, and greater systemic exposure, suggesting enhanced absorption and slower elimination in the freeze-dried formulation. Quercetin exhibited contrasting patterns. In the 50H group, quercetin showed an extended T1/2 despite lower Cmax and reduced AUC compared to FDH. Chlorogenic acid pharmacokinetics were comparable between the 50H and FDH groups. Scutellarin displayed the most pronounced processing-dependent differences. The 50H group has a substantially prolonged T1/2 and slightly elevated AUC. However, there is no significant difference detected among the metabolic fate of rutin, quercetin, and chlorogenic acid derived from the FDH and 50 °C groups.

## 4. Discussion

The drying rate of HCT was significantly influenced by hot-air temperature, confirming it as a critical determinant of drying kinetics. Elevated temperatures enhance thermal energy input and air convection, thereby accelerating moisture removal and reducing processing time [8,44]. Notably, moisture loss predominantly occurred during the falling-rate period), attributed to thermally driven moisture diffusion. The enhanced drying efficiency can be attributed to a transient temperature gradient between the material’s surface and interior, which drove internal moisture migration toward the evaporative surface [45]. In general, the effective moisture diffusivity ($D_{eff}$) increases with temperature [46]. However, in the present study, the highest $D_{eff}$ value was observed at 50 °C. This deviation from the expected trend may be attributed to the fact that, at elevated temperatures, the surface evaporation rate of moisture from HCT exceeds the internal moisture migration rate. This imbalance can lead to the formation of a hardened surface layer, which hinders further internal moisture diffusion and consequently results in a lower $D_{eff}$ [47].

The Aghbashlo model provided the most accurate description of HCT’s moisture dynamics under various drying conditions, successfully delineating critical transitions between drying stages. The activation energy (*Ea*) [48] served as an indicator of temperature sensitivity: a low *Ea* signified the dominance of external factors (such as temperature and airflow) in accelerating moisture evaporation, while a high *Ea* implied that the drying rate was limited by internal diffusion [49].

HCT contains diverse bioactive compounds, including organic acids, phenolic acids, flavonoids, terpenoids, and fatty acids, which contribute to its phytochemical properties, such as antioxidant, anti-inflammatory, antimicrobial, and antitumor activities [50,51]. LC-MS analysis revealed temperature-dependent variations in these phytochemicals across different drying methods. FD demonstrated superior preservation of chemical constituents, resulting in minimal compound degradation. In contrast, while hot-air drying (HAD) at 40 °C retained a similar range of compounds as FD, the appearance of new components indicated thermally induced transformations. At elevated temperatures, progressive compound loss became apparent. To illustrate, 50 °C HAD caused partial degradation, while 60 °C HAD led to significant phytochemical depletion. Quantitative analysis of four bioactive components confirmed superior retention in FD samples. Despite the reduced analyte content in HAD groups, temperature fluctuations between 40 and 60 °C had a minimal additional effect, indicating a potential thermal degradation threshold.

The quantitative analysis of total phenolic content followed a similar trend. Interestingly, the total flavonoid content in hot-air-dried (HAD) samples at 40 °C and 50 °C surpassed that of freeze-dried (FD) samples. This phenomenon may be attributed to the thermal degradation of cell wall structures, which facilitates the release of bound flavonoids through hydrolysis. The heat-induced transformation likely promotes the conversion of soluble flavonoids to insoluble bound forms, as well as the deglycosylation of flavonoid glycosides into their aglycone counterparts [52]. These structural modifications directly contributed to the observed variations in antioxidant capacity. Our findings demonstrate that freeze-drying exhibits superior efficacy in preserving the bioactive constituents of HCT. This conclusion was further substantiated by in vitro antioxidant activity assays. Within the hot-air-drying group, samples processed at 50 °C exhibited enhanced comprehensive antioxidant capacity, suggesting that optimal thermal conditions can effectively maintain the high antioxidant activity of phenolic compounds. Previous studies indicate that such effects may arise from synergistic interactions between non-enzymatic reactions and the inherent stability of phenolic compounds under controlled thermal conditions [53].

First, rapid surface moisture evaporation may induce case hardening, forming a dense outer layer that impedes the diffusion and extraction of internal bioactive compounds. Additionally, thermal degradation of cellular structures can lead to the entrapment or leakage of intracellular constituents [50]. Moreover, elevated temperatures accelerate non-enzymatic browning reactions, particularly the Maillard reaction, which generates melanoidins and contributes to the characteristic darkening of hot-air-dried samples [51,54]. Hot-air drying may also alter the composition and content of polyphenols due to peroxidation reactions, enzymatic browning, and Maillard reactions [55]. Finally, the intrinsic plant characteristics, such as tissue morphology and biochemical composition, further affect the drying kinetics and final product attributes [56].

Different drying conditions significantly influence the flavor profile of HCT. The acidity values across all experimental groups were below the tasteless threshold, with a gradual decline observed as the drying temperature increased. This trend suggests that elevated temperatures may promote the volatilization or thermal decomposition of acidic compounds during hot-air drying. Specifically, low-temperature drying appears better suited for preserving acidic components in HCT, whereas high-temperature conditions accelerate their degradation [57]. Moreover, bitterness, a dominant sensory attribute of HCT, exhibited notable variations among drying treatments. The 40 °C group displayed the highest bitterness intensity, while the 60 °C group registered the lowest. Interestingly, the FD and 50 °C groups demonstrated comparable bitterness levels, indicating a non-linear relationship between drying temperature and bitter compound retention. Furthermore, bitter aftertaste, referred to as a measure of lingering bitterness, was only detectable in the 40 °C group, suggesting that this moderate drying temperature may better preserve the persistence of bitter constituents. In terms of astringency, the 40 °C group also showed greater prominence. Comparative phytochemical analysis revealed that HCT dried at 40 °C retained the highest concentrations of polyphenols and unsaturated fatty acids. Given that most polyphenols contribute to bitterness and some elicit astringency [58], these findings align with the observed sensory outcomes.

The perception of bitterness in tannin-rich substances involves a neurophysiological pathway wherein interaction signals between tannins and salivary proteins are transmitted via neurons to the nucleus tractus solitarius (NTS). The NTS subsequently activates its gustatory region, ultimately relaying the signal to the brain’s taste cortex, where bitterness is consciously perceived [59]. Regarding olfactory characteristics, hot-air-dried HCT exhibits a more pronounced fishy odor compared to freeze-dried HCT. This difference can be attributed to the accumulation of volatile aldehydes and ketones—particularly decanoyl acetaldehyde—which are key contributors to fishy off-notes [60]. Analytical results confirm that hot-air drying induces compositional changes in HCT, notably increasing the diversity of aldehydes and ketones. This aligns with the observed intensification of fishy odor, whereas freeze-drying better preserves the original flavor profile by minimizing the generation of such volatile compounds. Comparative analysis of drying methods reveals distinct impacts on HCT’s flavor preservation and modification. Freeze-drying demonstrates superior efficacy in conserving HCT’s inherent flavor profile, particularly its characteristic spice and fruity notes. In contrast, hot-air drying tends to accentuate dry, woody flavor components, potentially through thermal degradation or oxidation processes. Notably, thermal processing at 60 °C induces significant flavor profile alterations, likely attributable to thermally driven chemical transformations, including the formation of novel alcohol and ester compounds. These findings establish a scientific foundation for process optimization, enabling targeted preservation or enhancement of specific flavor characteristics in HCT products.

Our comparative analysis revealed significant differences in bioactive compound preservation across drying methods. FDH demonstrated optimal retention of active constituents, while 50 °C drying achieved a favorable balance between drying efficiency (quantified by effective diffusivity, $D_{eff}$) and compound preservation. The 40 °C hot-air-drying samples showed suboptimal drying kinetics, whereas 60 °C conditions induced thermal degradation of thermolabile components, precluding their selection for subsequent antioxidant evaluation. Human HepG2 cells are widely used to study the regulatory effects of dietary compounds on hepatic antioxidant defense mechanisms. They are capable of absorbing and metabolizing bioactive substances such as flavonoids and alkaloids [61]. Moreover, liver mitochondria are the primary source of reactive oxygen species (ROS) [62]. These characteristics make HepG2 cells a suitable model for evaluating cellular antioxidant responses. Antioxidant activity was evaluated by measuring superoxide dismutase (SOD) activity, malondialdehyde (MDA) content, and intracellular ROS levels. SOD activity reflects the enzymatic conversion of superoxide radicals into less reactive species and represents a core component of the cellular antioxidant defense system. MDA is a marker of lipid peroxidation and cellular membrane damage, while ROS levels directly indicate oxidative stress intensity [63,64]. This comprehensive assessment strategy provided a robust framework for elucidating the cellular antioxidant mechanisms of HCT extracts under different processing conditions. The cell-based assays demonstrated that the FDH exhibited superior antioxidant efficacy by significantly preserving SOD activity, suppressing MDA accumulation, and scavenging excess ROS, possibly due to their high content of phenolic acids and flavonoids. While the extracts of the 50H group also effectively reduced ROS and MDA levels, its antioxidant capacity was comparatively weaker than FDH. These findings indicate that HCT extracts alleviate oxidative stress in HepG2 cells by enhancing endogenous antioxidant enzyme activity, reducing oxidative damage to membrane lipids, and decreasing intracellular ROS accumulation.

Pharmacokinetic analysis revealed significant processing-dependent variations in bioactive compound disposition. FDH exhibited superior pharmacokinetic profiles for rutin, quercetin, and chlorogenic acid, likely due to better preservation of these thermolabile compounds during lyophilization. In contrast, scutellarin demonstrated distinct pharmacokinetic behavior, with 50H showing significantly prolonged elimination T_1/2_, suggesting slower metabolic clearance. FDH yielded higher C_max_ values for scutellarin, aligning with the quantitative analysis showing better preservation in freeze-dried material; however, AUC differences between processing methods were less marked for this compound. Even though freeze-drying preserved the higher content of phenolic and flavonoid compounds in HCT compared with hot-air-drying HCT at 50 °C, there is no significant distinguished metabolic profiling of key bioactive compounds, including rutin, quercetin, and chlorogenic acid, between the FDH and 50H groups. That suggested that administration of hot-air drying HCT at 50 °C potentially exerts compatible antioxidant, anti-inflammatory, and cytoprotective effects in the host.

The differences in metabolites identified between the FDH and 50H groups suggest that the biological activity of Houttuynia cordata may be altered by processing. In the FDH group, the M1 metabolite tartaric acid (TA) not only mitigates hyperglycemia and hyperlipidemia [18] but also demonstrates potent antioxidant effects by scavenging free radicals and boosting SOD enzyme activity [65,66]. Additionally, the M18 metabolite hispidulin exhibits diverse biological activities. It is not only a promising anticancer agent but also possesses antioxidant, anti-inflammatory, and pro-apoptotic properties [67]. The M23 metabolite apigenin is a potent antioxidant that modulates Nrf-2/NF-κB/MAPK/PI3K/Akt signaling pathways and inhibits COX-2/NOS/XO oxidase activities [68,69]. Moreover, the proportion of fatty acid metabolites was relatively high in the 50H group. The M13 metabolite 3-hydroxybutyrate (3-HB), a key ketone body, exerts diverse therapeutic effects, including blood pressure reduction, myocardial ischemia protection, and cancer cell apoptosis induction via HDAC inhibition. Additionally, it alleviates neuroinflammation and oxidative stress in neurodegenerative diseases, serves as a vital energy source in diabetes, and plays a central role in the ketogenic diet (KD). However, its long-term elevation may lead to adverse effects such as cardiac fibrosis, liver inflammation, and gut microbiota imbalance [70,71]. In addition, the M15 metabolite 5-amino-2-hydroxybenzoic acid has antioxidant activity by scavenging free radicals via phenolic hydroxyl-derived H^+^/e^−^ donation [72]. The M28 metabolite known as the malonic acid inhibits NO, IL-6, and IL-1β production in LPS-induced BV2 microglia cells by blocking iNOS expression and downregulating the NF-κB and p38 MAPK pathways, suggesting potential neuroinflammation therapy [73]. It can also disrupt mitochondrial function by inhibiting SDH, leading to TCA cycle dysfunction and succinate-driven ROS accumulation via reverse electron transport [74,75,76]. The M37 metabolite 9-octadecenoic acid potentially enhances the immune function by boosting macrophage and neutrophil activity, reducing inflammation in autoimmune diseases. It can also lower cholesterol and blood pressure, promote wound healing, inhibit cancer cell growth, improve insulin sensitivity for metabolic disorders, and regulate membrane fluidity via regulating NF-κB, p38 MAPK, and G protein signaling pathways [77]. The results of metabolic profiling suggested distinct bioactivity patterns. The FDH group was enriched in phenolic acid and flavonoid metabolites, which may explain their superior antioxidant, anti-inflammatory, and cytoprotective effects. In contrast, the 50H group demonstrated increased fatty acid and phenolic acid metabolites, indicating stronger modulation of energy and lipid metabolism pathways. These metabolic profiles provide a mechanistic basis for understanding the impact of drying processes on the functional quality of HCT. The differences in the biological activities of these metabolites indicate that processing can alter the composition and content of Houttuynia cordata metabolites, thereby affecting their overall bioactivity. Such changes may enhance or diminish specific effects, ultimately influencing its functional performance. These hypotheses, however, require further in vivo validation. Our study establishes the first comprehensive framework for optimizing both quality and efficiency in the processing of HCT, revealing distinct metabolic transformation pathways induced by different drying methods. This work fills a critical gap in HCT research and contributes to the standardization of drying processes as well as improved energy efficiency in its industrial application.

## 5. Conclusions

This study demonstrates that drying methods significantly influence the phytochemical composition, bioactivity, and flavor characteristics of *Houttuynia cordata Thunb.* Freeze-drying (FD) most effectively preserves antioxidant phenolic acids and flavonoids, while hot-air drying at 50 °C achieves a balance between drying efficiency and partial retention of bioactive compounds. The observed trade-offs between drying temperature, compound stability, and functional properties underscore the need to optimize drying processes based on application-specific requirements. Metabolic profiling of rats administered freeze-dried and hot-air-dried HCT extracts revealed distinct bioactivity patterns. The FD group exhibited higher levels of phenolic acid and flavonoid metabolites, aligning with their superior antioxidant, anti-inflammatory, and cytoprotective effects. In contrast, the 50 °C hot-air-dried (50H) group showed increased fatty acid and phenolic acid metabolites, suggesting enhanced modulation of energy and lipid metabolism pathways. Notably, no significant differences were observed in the plasma concentrations of key bioactive compounds (e.g., rutin, quercetin, chlorogenic acid, and scutellarin) between the FD and 50H treatments, indicating that hot-air drying at 50 °C may yield comparable in vivo antioxidant efficacy. These findings provide mechanistic insights into how drying processes alter the functional quality of herbaceous plants and their subsequent health impacts. Collectively, this study provides empirical support for customizing HCT processing parameters to optimize functional outcomes while establishing a basis for subsequent investigations into processing methodology improvements.

## Figures and Tables

**Figure 1 foods-14-01962-f001:**
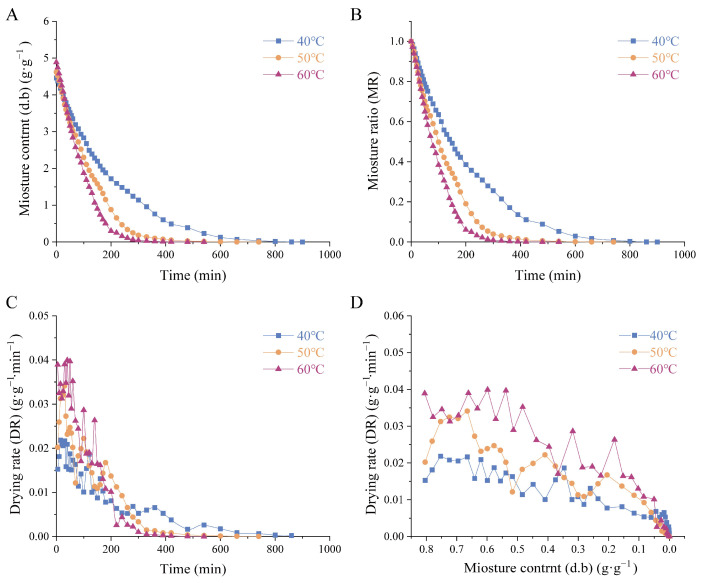
Drying kinetic curves of *Houttuynia cordata Thunb.* (**A**) dry basis moisture content of HCT, (**B**) moisture ratio curve of HCT. (**C**) drying rate with the wet basis moisture content of the curve, (**D**) drying rate curves of HCT at different drying temperatures. 40 °C/50 °C/60 °C: hot-air drying at 40 °C, 50 °C, and 60 °C HCT respectively.

**Figure 2 foods-14-01962-f002:**
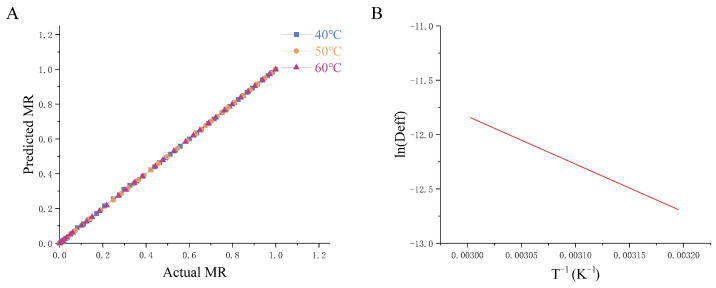
Relationship between hot-air thin-layer drying temperature and effective diffusion coefficient of *Houttuynia cordata Thunb.* drying by hot air. (**A**) The predicted *MR* by the Aghabashlo model vs. Actual *MR*, (**B**) Arrhenius type relationship between effective diffusivity and temperature. 40 °C/50 °C/60 °C: hot-air drying at 40 °C, 50 °C, and 60 °C HCT respectively.

**Figure 3 foods-14-01962-f003:**
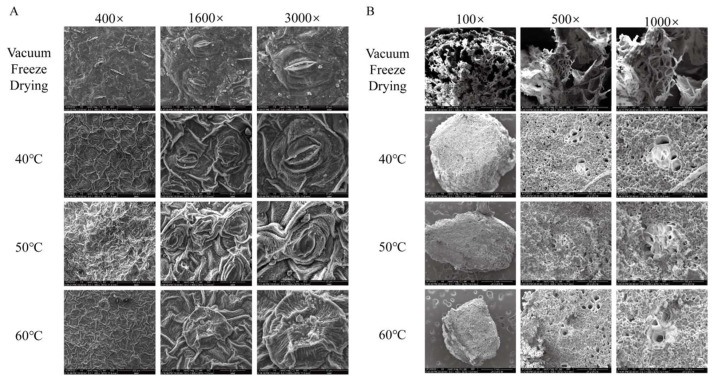
Microstructure diagrams of *Houttuynia cordata Thunb.* (**A**) *Houttuynia cordata Thunb.* Leaves, (**B**) *Houttuynia cordata Thunb.* Stems were dried under vacuum freeze-drying conditions (freeze-drying sample) at 40 °C hot-air drying, 50 °C hot-air drying, and 60 °C hot-air drying. 100×: 100 times magnification, 400×: 400 times magnification, 1600×: 1600 times magnification, 3000×: 3000 times magnification.

**Figure 4 foods-14-01962-f004:**
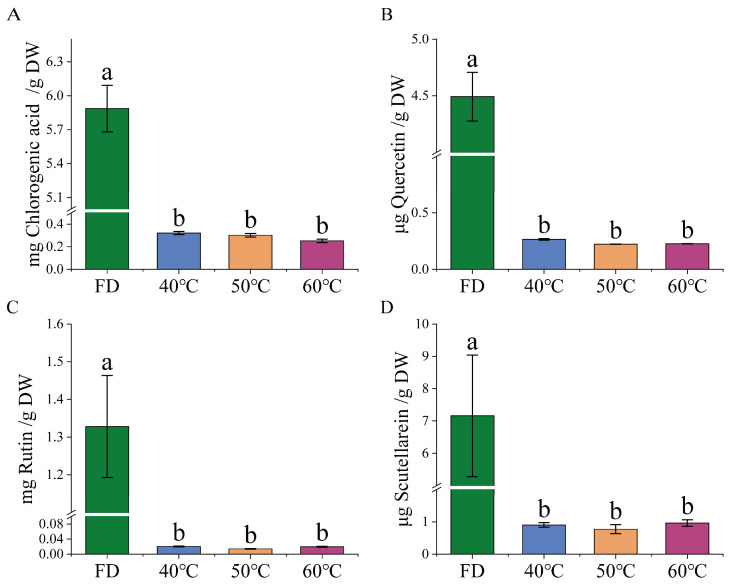
Determination of *Houttuynia cordata Thunb.* content in freeze-drying (FD), 40 °C hot-air drying HCT extracts (40 °C), 50 °C hot-air drying HCT extracts (50 °C) and 60 °C hot-air drying HCT extracts (60 °C), (*n* = 3). (**A**) chlorogenic acid, (**B**) quercetin, (**C**) rutin, and (**D**) scutellarin. Groups without common letters are significantly different (*p* ≤ 0.05).

**Figure 5 foods-14-01962-f005:**
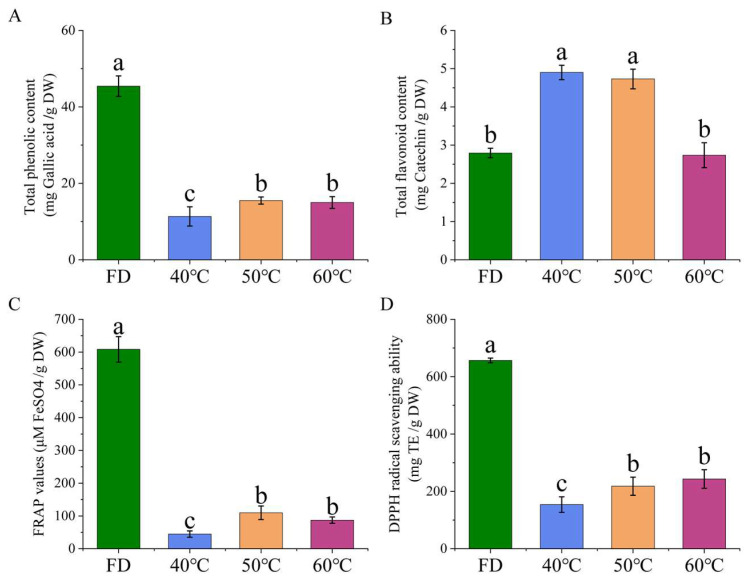
Phytochemical contents and antioxidant activities in *Houttuynia cordata Thunb.* extracts under different drying conditions (*n* = 6). (**A**) Total phenolic content; (**B**) total flavonoid content; (**C**) antioxidant capacity of FRAP; (**D**) DPPH scavenging capacity. FD: freeze-drying; 40 °C, 50 °C, 60 °C: hot-air drying at 40 °C, 50 °C, and 60 °C respectively. Values without common letters indicate significant differences (*p* ≤ 0.05).

**Figure 6 foods-14-01962-f006:**
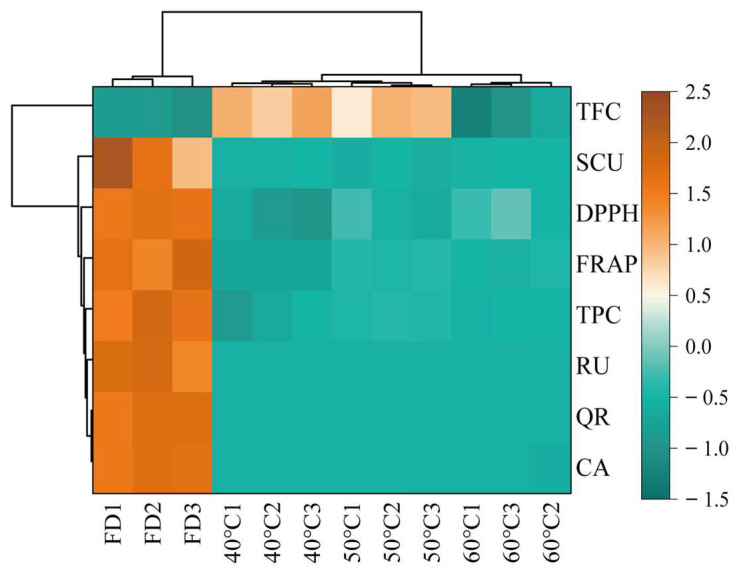
Heatmap visualization. Rows represent bioactive compounds (TFC, SCU, DPPH, FRAP, TPC, RU, QR, CA), while columns represent samples grouped by processing methods (FD: freeze-drying HCT; 40 °C/50 °C/60 °C: hot-air drying at 40 °C, 50 °C, and 60 °C HCT respectively). Data are normalized by row Z-scores (color scale: brown = high abundance [2.5], green = low abundance [−1.5]).

**Figure 7 foods-14-01962-f007:**
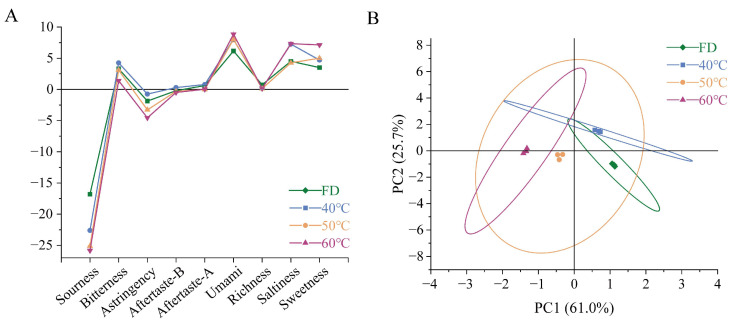
Comparative analysis of flavor profiles in Houttuynia cordata under different drying conditions. (**A**) Test results of 9 flavors, (**B**) PCA score plot. FD: freeze-drying HCT; 40 °C/50 °C/60 °C: hot-air drying at 40 °C, 50 °C, and 60 °C HCT respectively.

**Figure 8 foods-14-01962-f008:**
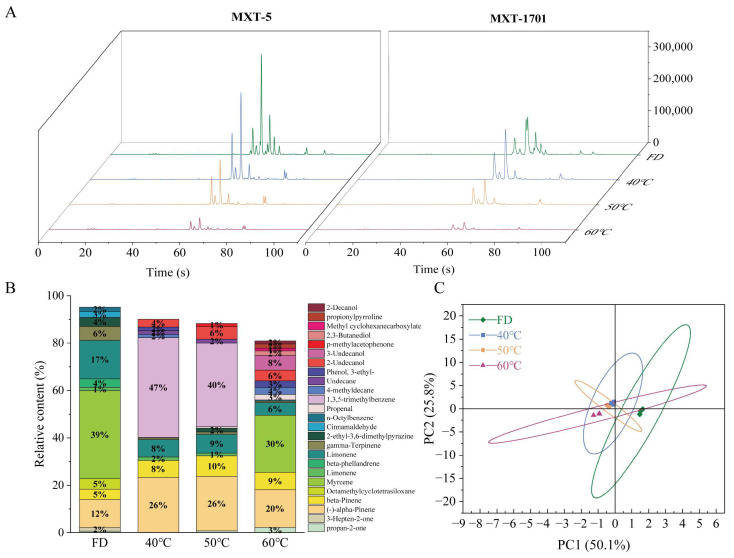
Flavor profile analysis of Houttuynia cordata under different drying conditions using electronic nose technology (**A**) Total ion chromatogram, (**B**) Relative abundance of volatile compounds, (**C**) PCA score plot. FD: freeze-drying HCT; 40 °C/50 °C/60 °C: hot-air drying at 40 °C, 50 °C, and 60 °C HCT respectively.

**Figure 9 foods-14-01962-f009:**
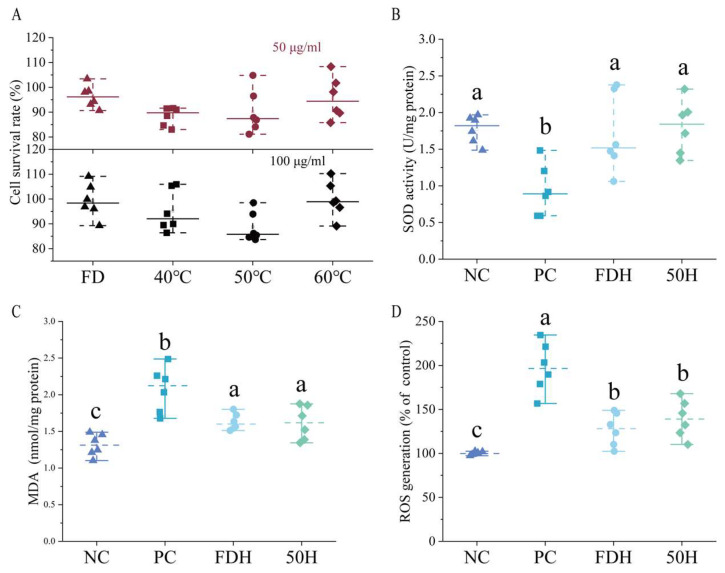
Protective effects of Houttuynia cordata extract against H_2_O_2_-induced oxidative stress in HepG2 cells (*n* = 6). (**A**) Cell viability treated with 50 μg/mL and 100 μg/mL of HCT extract, (**B**) SOD activity, (**C**) MDA content, and (**D**) ROS levels. Different lowercase letters above bars indicate statistically significant differences among groups (*p* < 0.05). NC: negative control group; PC: positive control group; FDH: freeze-dried HCT group, 50H: 50 °C hot-air drying HCT group.

**Figure 10 foods-14-01962-f010:**
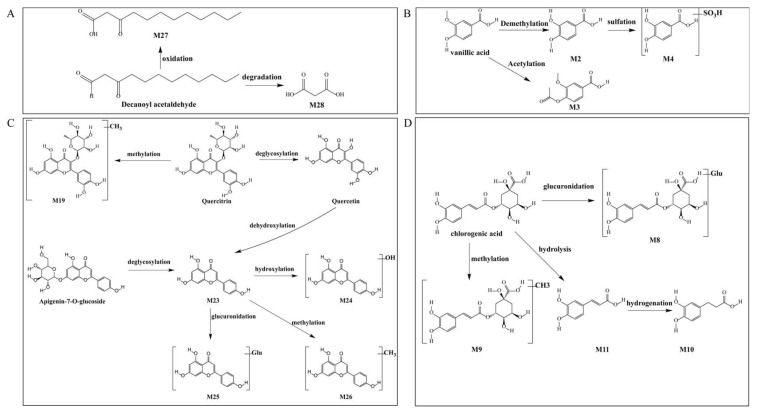
Metabolic pathways of bioactive compounds in rats: (**A**) decanoyl acetaldehyde (**B**) quercetin and apigenin (**C**) chlorogenic acid (**D**) vanillic acid.

**Figure 11 foods-14-01962-f011:**
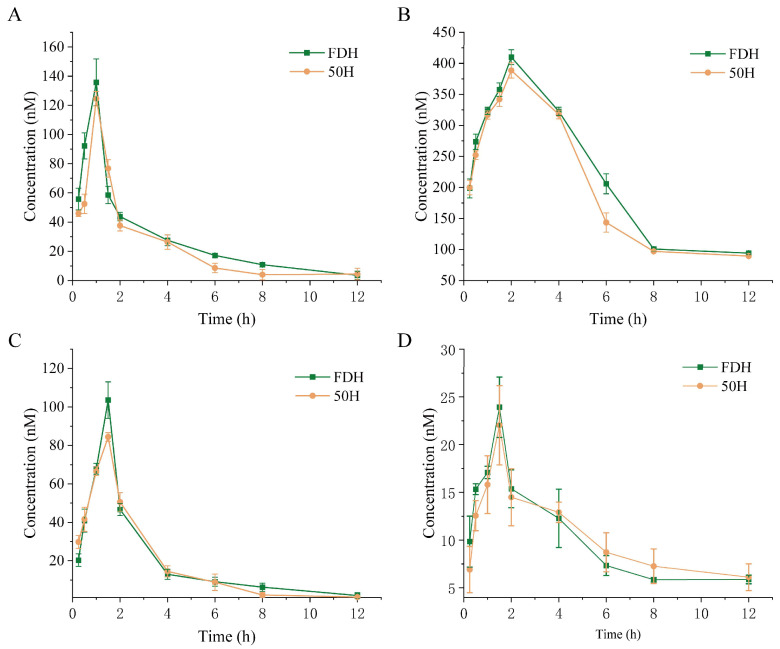
Blood plasma concentration–time curves of (**A**) rutin, (**B**) quercetin, (**C**) chlorogenic acid, and (**D**) scutellarin in rats after single-dose administration of FDH and 50H formulations. FDH: freeze-dried HCT group, 50H: 50 °C hot-air drying HCT group.

**Table 1 foods-14-01962-t001:** The typical thin-layer models selected for *Houttuynia cordata Thunb*. drying curve.

Model NO.	Model Name	Model Equation	References
1	Lewis/Newton	${MR=exp}^{\left(-kt\right)}$	[19]
2	Page	${MR=exp}^{\left({-kt}^{n}\right)}$	[20]
3	Henderson/Pabis	${MR=a\;exp}^{\left(-kt\right)}$	[21]
4	Two-term	$MR={a\;exp}^{\left(-kt\right)}+{c\;exp}^{\left(-gt\right)}$	[22]
5	Two-term Exponential	$MR={a\;exp}^{\left(-kt\right)}+{\left(1-a\right)exp}^{\left(-kat\right)}$	[23]
6	Logarithmic	$MR=a\;{exp}^{\left(-kt\right)}+c$	[24]
7	Midilli	$MR=a\;\exp\left(-kt^{n}\right)+bt$	[25]
8	Aghabashlo	$MR=\exp\left[-kt/\left(1+gt\right)\right]$	[26]
9	Wang and Singh	$MR=1+bt+at^{2}$	[27]
10	Silva	$\mathrm{MR}=exp(-at-bt^{\frac{1}{2}})$	[28]

Note: *t*. Drying time (s); *a*, *k*, *b*, *g*, *n*, *c*. Model coefficients.

**Table 2 foods-14-01962-t002:** The substance changes of *Houttuynia cordata Thunb.* after drying were identified by UPLC-LTQ-Orbitrap-MS.

Peak	t_R_ (min)	Formula	[M − H]^−^ (*m*/*z*)	Fragments (MS^2^)	FD	40 °C	50 °C	60 °C	Tentative Identification
1	1.11	C_4_H_6_O_5_	133.01	115.19, 71.07	+	+	+	+	malic acid
2	1.12	C_6_H_8_O_7_	191.02	111.04	+	+	+	—	vanillic acid
3	2.02	C_8_H_8_O_4_	167.04	149.13, 123.06	+	+	+	+	n6-succinyl adenosine
4	2.15	C_14_H_17_N_5_O_8_	382.10	206.10, 162.105, 134.04	—	+	+	+	vannilic acid-4-o-β-l-rhamnoside
5	2.76	C_14_H_18_O_8_	313.09	167.27, 123.14, 121.24	+	+	+	+	5-(β-d-glucopyranosyloxy)-2-hydroxybenzoic acid
6	2.95	C_13_H_16_O_9_	315.11	153.07	+	+	+	+	quinic acid
7	3.06	C_7_H_12_O_6_	191.06	173.23, 171.04, 127.00, 111.05	+	+	+	+	isovanillin
8	3.30	C_8_H_8_O_3_	151.04	91.30	+	+	+	+	chlorogenic acid
9	3.51	C_16_H_18_O_9_	353.09	191.08, 179.06, 135.04	+	+	+	+	caffeic acid-O-hexoside
10	5.46	C_15_H_18_O_9_	341.09	179.06, 135.04	+	+	—	+	methyl 4-acetoxy-3-hydroxybutanoate
11	5.32	C_7_H_12_O_5_	175.06	115.04, 85.15	+	+	+	+	kaempferol-3-rutinoside
12	7.40	C_27_H_30_O_15_	593.15	285.01	—	+	+	+	5-acetamido-2-(2-carboxyethyl) benzoic acid
13	9.69	C_12_H_13_NO_5_	250.07	206.07, 164.93	+	—	+	—	Houttuynoid A
14	9.82	C_33_H_38_O_13_	641.17	479.17, 461.32	—	+	—	—	5-O-p-Coumaroylquinic acid
15	9.96	C_16_H_18_O_8_	337.18	191.08	+	+	+	—	3,4-Dihydroxybenzoic acid
16	10.25	C_7_H_6_O_4_	153.02	109.22, 91.34	+	+	—	—	rutin
17	11.72	C_27_H_30_O_16_	609.15	300.97, 271.05	+	+	+	+	hispidulin-O-hexoside
18	12.05	C_22_H_22_O_11_	461.07	299.06, 283.06	+	+	+	+	quercetin-O-hexoside
19	12.29	C_21_H_20_O_12_	463.09	301.06, 300.06	+	+	+	+	7-O-methylmangiferin
20	12.68	C_20_H_20_O_11_	435.09	315.10	+	+	+	+	kaempferol-O-coumaroylhexoside
21	13.40	C_30_H_26_O_13_	593.15	447.20, 285.08, 255.12	+	+	+	—	quercitrin
22	14.25	C_21_H_20_O_11_	447.09	401.26, 301.11, 271.29	+	+	+	+	isorhamnetin-3-glucopyranoside
23	14.57	C_22_H_22_O_12_	477.11	315.03	+	+	—	—	apigenin-7-O-glucoside
24	16.37	C_21_H_20_O_10_	431.10	285.08	+	+	+	—	scutellarein
25	16.75	C_15_H_10_O_6_	285.04	151.19	+	+	+	+	quercetin
26	19.87	C_15_H_10_O_7_	301.04	178.98, 151.04	+	+	+	+	ouercetol
27	20.04	C_15_H_10_O_7_	301.04	273.15, 151.03	—	+	+	+	dihydroisoferulic acid-3-o-glucuronide
28	20.53	C_16_H_20_O_10_	371.11	191.09, 134.08	+	+	—	—	decanoyl acetaldehyde
29	28.78	C_12_H_22_O_2_	197.81	179.13, 169.09, 154.06, 135.91	+	+	+	+	10-hydroxydecanoic
30	29.55	C_10_H_20_O_3_	187.13	141.10	+	+	+	+	(6e,8z)-18-hydroxy-5-oxo-6,8-octadecadienoic acid or isomer
31	28.61	C_18_H_30_O_4_	309.21	291.26, 221.18	—	+	+	+	decanoic acid
32	29.16	C_18_H_28_O_4_	307.19	235.15, 125.25	—	+	+	+	12,13,17-trihydroxyoctadec-9-enoic acid
33	29.18	C_18_H_34_O_5_	329.23	229.16, 211.27, 193.09, 171.08	—	+	+	+	10-hydroxydecanoic acid
34	30.08	C_13_H_24_O_3_	227.13	183.16	—	+	—	—	l-menthyl lactate
35	31.44	C_17_H_28_O_5_	311.22	293.32	—	+	+	+	arteether
36	32.22	C_18_H_30_O_3_	293.21	275.25, 223.12, 205.23	—	+	+	+	12-hydroxy-9-octadecenoic acid
37	32.41	C_18_H_36_O_4_	315.25	297.25, 171.26	—	+	—	—	9,12-dihydroxyoctadecanoic acid

+: Indicates detected by UPLC-LTQ-Orbitrap-MS; —: indicates not detected by UPLC-LTQ-Orbitrap-MS. FD: freeze-drying HCT; 40 °C/50 °C/60 °C: hot-air drying at 40 °C, 50 °C, and 60 °C HCT respectively.

**Table 3 foods-14-01962-t003:** Characterization of rat metabolites of *Houttuynia cordata Thunb.* extracts by UPLC-LTQ-Qrbitrap-MS^2^.

Peak	t_R_ (min)	Formula	[M − H]^−^ (*m*/*z*)	Fragments (MS^2^)	Serum		Identification	Original Compounds
					50H	FDH		
M1	39.41	C_4_H_6_O_6_	149.0107	74.9963, 59.9747	—	+	Hydroxylation	malic acid
M2	4.31	C_7_H_6_O_4_	153.0237	96.9634, 79.9619	+	+	Demethylation	vanillic acid
M3	7.92	C_10_H_10_O_5_	209.04566	165.056	+	+	Acetylation	vanillic acid
M4	1.77	C_7_H_6_O_7_S	232.97405	160.8908	+	+	Demethylation Sulfation	vanillic acid
M5	15.8	C_7_H_12_O_7_	207.05201	162.8414	—	+	Hydroxylation	Quinic acid
M6	1.14	C_8_H_8_O_4_	167.02224	167.02224	+	—	Hydroxylation	isovanillin
M7	3.8	C_14_H_16_O_9_	327.0712	283.0752	+	+	Glucuronidation	isovanillin
M8	30.44	C_22_H_26_O_15_	529.11817	200.8584	—	+	Glucuronidation	chlorogenic acid
M9	1.08	C_17_H_20_O_9_	367.10454	307.0841, 157.0321	+	—	Methylation	chlorogenic acid
M10	7.92	C_9_H_10_O_4_	181.05145	136.0184, 93.0297	+	+	Hydrolysis	chlorogenic acid
M11	9.75	C_9_H_8_O_4_	179.03513	143.8653, 134.9881	+	+	Hydrolysis	chlorogenic acid
M12	7.11	C_12_H_19_O_11_	338.08691	162.0569	+	+	Hydrolysis Glucuronidation	methyl 4-acetoxy-3-hydroxybutanoate
M13	2.13	C_4_H_8_O_3_	103.0427	59.0217	+	—	Hydrolysis	methyl 4-acetoxy-3-hydroxybutanoate
M14	8.21	C_10_H_11_NO_4_	208.06233	165.0614, 87.9269	+	+	Hydrolysis	5-acetamido-2-(2-carboxyethyl) benzoic acid
M15	15.08	C_7_H_7_NO_3_	152.03668	122.0381	+	-	Hydrolysis	5-acetamido-2-(2-carboxyethyl) benzoic acid
M16	9.54	C_9_H_8_O_3_	163.04073	147.8876, 119.0519	+	+	Hydrolysis	5-O-p-Coumaroylquinic acid
M17	11.13	C_7_H_6_O_3_	137.02574	93.0384	+	+	Hydrolysis	5-O-p-Coumaroylquinic acid
M18	15.83	C_16_H_12_O_6_	299.05479	284.0314	—	+	Hydrolysis	Hispidulin-O-hexoside
M19	15.56	C_22_H_22_O_11_	461.10722	285.0726	+	+	Methylation	Quercitrin
M20	12.49	C_21_H_22_O_11_	449.10576	273.0743	+	+	Methylation	7-O-methylmangiferin
M21	13.88	C_22_H_19_O_15_	522.07036	431.0637, 332.9840	—	+	Hydroxylation	isorhamnetin-3-glucopyranoside
M22	17.41	C_23_H_24_O_12_	491.11772	401.08	—	+	Methylation	isorhamnetin-3-glucopyranoside
M23	15.32	C_15_H_10_O_5_	269.04461	269.0444	—	+	Hydrolysis	Apigenin-7-O-glucoside
M24	13.49	C_15_H_10_O_6_	285.03853	202.8602, 150.9995, 133.0314	—	+	Hydrolysis, Hydroxylation	Apigenin-7-O-glucoside
M25	9.61	C_21_H_18_O_11_	445.07596	269.0446	+	+	Hydrolysis Glucuronidation	Apigenin-7-O-glucoside
M26	2.58	C_16_H_12_O_5_	283.06769	151.0272	+	+	Hydrolysis, Methylation	Apigenin-7-O-glucoside
M27	20.73	C_12_H_22_O_3_	213.14978	59.0198	+	+	Hydroxylation	decanoyl acetaldehyde
M28	1.2	C_3_H_4_O_4_	103.00603	59.0218	+	-	Hydrolysis	decanoyl acetaldehyde
M29	13.65	C_10_H_18_O_4_	201.1137	183.1047, 138.1147	+	+	Hydroxylation	10-hydroxydecanoic
M30	27.6	C_18_H_30_O_5_	325.18368	183.0123	+	—	Hydroxylation	(6e,8z)-18-hydroxy-5-oxo-6,8-octadecadienoic acid
M31	19.59	C_25_H_38_O_10_	497.23737	321.206	+	—	Hydroxylation Glucuronidation	(6e,8z)-18-hydroxy-5-oxo-6,8-octadecadienoic acid
M32	6.55	C_20_H_35_O_6_N	366.23453	304.2383	+	—	Glycine binding	(6e,8z)-18-hydroxy-5-oxo-6,8-octadecadienoic acid
M33	2.94	C_9_H_10_O_5_	197.04889	162.8403	+	—	Hydrolysis	arteether
M34	20.97	C_18_H_28_O_4_	307.19112	289.1816, 191.1061	+	—	Hydroxylation	12-hydroxy-9-octadecenoic acid
M35	16.95	C_18_H_32_O_5_	327.2164	171.0989	+	+	Hydroxylation	12-hydroxy-9-octadecenoic acid
M36	30.4	C_18_H_32_O_2_	279.23254	279.2324	+	+	Hydrolysis	12-hydroxy-9-octadecenoic acid
M37	28.38	C_18_H_30_O_2_	277.21662	165.0575	+	+	Restore	12-hydroxy-9-octadecenoic acid
M38	22.57	C_24_H_44_O_10_	491.28011	491.2839	+	—	Glucuronidation	9,12-dihydroxyoctadecanoic acid
M39	20.32	C_19_H_38_O_4_	329.26843	329.2694	+	+	Methylation	9,12-dihydroxyoctadecanoic acid
M40	22.12	C_18_H_34_O_4_	313.23795	201.1134	+	+	Hydroxylation	9,12-dihydroxyoctadecanoic acid

+: detected; —: undetected. FDH: freeze-dried HCT group, 50H: 50 °C hot-air drying HCT group.

**Table 4 foods-14-01962-t004:** Pharmacokinetic parameters of bioactive compounds in differently processed HCT.

Compound		T_1/2_ (h)	T_max_ (h)	C_max_ (nM)	AUC (h × nM)
Rutin	50H	2.04 ± 0.50	1.0	124.55 ± 4.95	277.12 ± 12.26
	FDH	2.61 ± 1.23	1.0	135.77 ± 16.02	328.20 ± 5.68
Quercetin	50H	7.28 ± 4.47	2.0	377.93 ± 21.44	2320.45 ± 57.37
	FDH	4.46 ± 0.14	2.0	410.07 ± 12.00	2550.76 ± 17.03
Chlorogenic acid	50H	2.05 ± 0.62	1.5	84.39 ± 2.26	221.69 ± 22.46
	FDH	2.25 ± 0.70	1.5	103.56 ± 9.47	231.19 ± 7.87
Scutellarin	50H	9.26 ± 4.35	1.5	22.03 ± 4.15	120.56 ± 4.62
	FDH	6.91 ± 0.65	1.5	23.92 ± 3.17	115.99 ± 7.02

T_1/2_: Terminal half-life, T_max_: Time to peak concentration, C_max_: Peak plasma concentration, AUC: Area under the concentration–time curve.

## Data Availability

The raw data supporting the findings of this study are not publicly available as the data are currently under review for patent protection. As the patent application is in the confidential examination stage and has not yet been disclosed, the data cannot be shared at this time. Reasonable requests for further information can be directed to the corresponding author, subject to appropriate confidentiality considerations.

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
