# Peer review of "Comprehensive Evaluation of the Effects of Hot Air Drying Temperature on the Chemical Composition, Flavor Characteristics and Biological Activity of Houttuynia cordata Thunb."

_foods, 2025, doi:10.3390/foods14111962_

Round 1

Reviewer 1 Report

Comments and Suggestions for Authors

This manuscript contributes to the significant subject of hot-air thin layer drying of Houttuynia cordata Thunb, a well-known food and medicinal plant mainly in Asia. The topic is worth investigating indeed, and the authors describe all the aspects (the microstructure, bioactive compound, volatile compound, and overall retention) of the drying process. This could be interesting to researchers from the food processing and preservation field. However, to strengthen the study's impact, the following aspects could be refined.

No

Section

Comments

1.

Introduction

Introduction

Although the introduction gives an informative overview of Houttuynia cordata and drying methods, it lacks a clearer rationale regarding the choice of hot air thin-layer drying. Please explain why this method is more advantageous than other methods like microwave, freeze-drying, or infrared drying, citing literature or providing a technical explanation.

Additionally, why did the authors compare hot air drying with freeze drying in the results section? Please add additional information regarding to it.

Lines 75–80, the authors state that “there are many evidences that the hot air thin-layer drying has been successfully applied to process various plant food materials, particularly vegetables and fruits.” I would suggest adding specific studies or providing examples that illustrate successful applications of hot air thin-layer drying to support that sentence.

Since in vivo pharmacokinetic and metabolic experiments using rats are part of the study (materials and methods section, Line 276-289), I would suggest that the Introduction and study objectives should clearly indicate this aspect. This would help readers understand how the animal study fits into the overall framework of the research as well as provide a more coherent rationale for the experimental design.

2

Materials and methods

Sample identification was done by a professor (Line 94-95), but lacks its clarity to which he was identified.

The Materials and Methods section should be rechecked and reviewed to make sure that all key procedures are properly referenced and justified.

Eg.  Section 2.4 ("Preparation of HCT Extracts", lines 174–177), the concentration of ethanol used, solid-to-liquid ratio, and extraction time have been described in detail but not supported with a reference.

The sentence from (Line 211) 'All analyses were performed with six replicates (n = 3)' is unclear and somewhat confusing. I would suggest that the authors clarify their replication strategy.

I recommend that the authors need to provide detailed descriptions of the sample preparation, operating conditions, and equipment used for hot air drying and freeze drying in the Materials and Methods section, as these methodologies are discussed in the Results section.

3.

Results and discussion

The mention of HPLC-LTQ-Orbitrap-MS (Line 375) should be moved to the Materials and Methods section, as analytical instrument details are more appropriate there than in the Results section.

The title in Lines 355–356 suggests a detailed examination of how varying hot air-drying temperatures affect the chemical composition and antioxidant activity. However, the paragraph in lines 357–371 discusses the advantages of freeze-drying as a preservation method, emphasizing its effectiveness in maintaining bioactive compounds and ensuring sample stability.

Therefore, I recommend relocating this paragraph to a more appropriate section and revising the current section (3.3) to focus on the effects of different hot air drying temperatures.

The titles of Sections 3.3 (Line 355) and 3.4 (Line 468) are identical. Please review and revise them to ensure each section has a distinct and appropriate heading.

4.

Conclusion

I recommend incorporating practical applications derived from the current results for industrial purposes and please highlight the novelty of this research.

5.

Overall

I would recommend reviewing the formatting of all scientific names throughout the manuscript to ensure they are consistently italicized.

Please make sure the Introduction and Materials and Methods sections are closely aligned with the Results section. The Introduction should mention the research objectives and rationale, while the Materials and Methods section should provide a detailed description of the experimental procedures, including sample preparation, operating conditions, equipment used, and analytical methods, with appropriate references.

Author Response

Comments 1: Although the introduction gives an informative overview of Houttuynia cordata and drying methods, it lacks a clearer rationale regarding the choice of hot air thin-layer drying. Please explain why this method is more advantageous than other methods like microwave, freeze-drying, or infrared drying, citing literature or providing a technical explanation.

Response 1: We sincerely appreciate the reviewer's insightful suggestion to strengthen the rationale for our methodological approach. We agree that providing a clearer justification for selecting hot air thin-layer drying will enhance the clarity of our study. We have added the technical explanation in the Introduction section (Lines 79-81, highlighted in light yellow).

Comments 2: Additionally, why did the authors compare hot air drying with freeze drying in the results section? Please add additional information regarding to it

Response 2: We sincerely appreciate the reviewer's valuable suggestion to clarify our rationale for comparing hot air drying with freeze-drying. We agree that this important methodological choice requires more explicit justification. We have expanded the explanation in the Results section (Lines 378–393, highlighted in light yellow)

Comments 3: Lines 75–80, the authors state that “there are many evidences that the hot air thin-layer drying has been successfully applied to process various plant food materials, particularly vegetables and fruits.” I would suggest adding specific studies or providing examples that illustrate successful applications of hot air thin-layer drying to support that sentence.

Response 3: We sincerely appreciate the reviewer's constructive suggestion to strengthen our evidence regarding hot air thin-layer drying applications. We agree that specific examples will better support our methodological choice. We have supplemented the text in the Introduction section (Lines 86-88, highlighted in yellow).

Comments 4: Since in vivo pharmacokinetic and metabolic experiments using rats are part of the study (materials and methods section, Line 276-289), I would suggest that the Introduction and study objectives should clearly indicate this aspect. This would help readers understand how the animal study fits into the overall framework of the research as well as provide a more coherent rationale for the experimental design.

Response 4: We sincerely appreciate the reviewer’s valuable suggestion to better integrate the in vivo pharmacokinetic study into the manuscript’s framework. We agree that clarifying this aspect in the Introduction will improve the coherence of our experimental design. To enhance clarity, we have revised the Introduction (Lines 97–110, highlighted in yellow) to explicitly state the research objectives, particularly the inclusion of in vivo metabolic experiments.

Comments 5: Sample identification was done by a professor (Line 94-95), but lacks its clarity to which he was identified.

Response 5: Thank you for your valuable comment. We agree that the original description lacked clarity regarding the identification process. We have revised the text to clearly state that Houttuynia cordata (HCT) was identified by Professor Jinlian Zhang from Jiangxi University of Traditional Chinese Medicine (Nanchang, China), based on its morphological characteristics, including the characteristic fishy odor, bitter taste, green upper surface, and purplish-red lower surface. This revision has been made in the Materials and Methods section on Lines 116121 of the revised manuscript.

Comments 6: The Materials and Methods section should be rechecked and reviewed to make sure that all key procedures are properly referenced and justified. Eg.  Section 2.4 ("Preparation of HCT Extracts", lines 174–177), the concentration of ethanol used, solid-to-liquid ratio, and extraction time have been described in detail but not supported with a reference.

Response 6: Thank you for your valuable comment. We agree with the reviewer that all key experimental procedures should be properly referenced to enhance the credibility and reproducibility of the study. Accordingly, we have added a supporting reference [73] for the extraction parameters described in Section 2.4 Preparation of HCT Extracts. The ethanol concentration, solid-to-liquid ratio, and extraction time are now justified with relevant literature. This citation has been inserted on Line 213, and has been highlighted in yellow in the revised manuscript.

Comments 7: The sentence from (Line 211) 'All analyses were performed with six replicates (n = 3)' is unclear and somewhat confusing. I would suggest that the authors clarify their replication strategy.

Response 7: Thank you for your helpful comment. We agree that the original sentence was unclear. To address this, we have revised the description to more accurately reflect our replication strategy. Specifically, we clarified that the LC-MS quantification experiments were conducted in three independent replicates. This correction enhances the clarity and reproducibility of our methodology. The revised text can be found on Line 246 of the updated manuscript and has been highlighted in yellow for easy reference. 

Comments 8: I recommend that the authors need to provide detailed descriptions of the sample preparation, operating conditions, and equipment used for hot air drying and freeze drying in the Materials and Methods section, as these methodologies are discussed in the Results section.

Response 8: We sincerely appreciate the reviewer's insightful comment regarding the need for a more precise description of the drying process. We fully agree that such technical details are essential for ensuring experimental reproducibility. Accordingly, we have revised the Materials and Methods section to include comprehensive specifications for both the hot-air drying and freeze-drying procedures, such as drying temperatures, durations, sample thickness, and equipment models. All modifications have been highlighted in yellow in the revised manuscript.  These changes can be found on Lines 124–137.

Comments 9: The mention of HPLC-LTQ-Orbitrap-MS (Line 375) should be moved to the Materials and Methods section, as analytical instrument details are more appropriate there than in the Results section.

Response 9: We sincerely appreciate the reviewer's valuable suggestion to improve the methodological clarity of our drying processes. We fully agree that consolidating all technical details in the Materials and Methods section enhances manuscript organization. The revised content can be found on Line 216 of the updated manuscript and has been highlighted in yellow for your convenience.

Comments 10: The title in Lines 355–356 suggests a detailed examination of how varying hot air-drying temperatures affect the chemical composition and antioxidant activity. However, the paragraph in lines 357–371 discusses the advantages of freeze-drying as a preservation method, emphasizing its effectiveness in maintaining bioactive compounds and ensuring sample stability. Therefore, I recommend relocating this paragraph to a more appropriate section and revising the current section (3.3) to focus on the effects of different hot air drying temperatures.

Response 10: Thank you for your insightful comment. We fully agree that the paragraph originally located in Lines 357–371 was better suited for a comparative discussion rather than within a section dedicated to hot-air drying temperature effects. Accordingly, we have revised Section 3.3 to more directly focus on the influence of different hot-air drying temperatures on the chemical composition and antioxidant activity of Houttuynia cordata. Additionally, we relocated the content related to freeze-drying to a more appropriate section and revised it to emphasize its relevance in comparison with hot-air drying. These changes improve the coherence and alignment between the section title and content.  The revised and relocated content can be found on Lines 393-408, and all modifications have been highlighted in yellow.

Comments 11: The titles of Sections 3.3 (Line 355) and 3.4 (Line 468) are identical. Please review and revise them to ensure each section has a distinct and appropriate heading.

Response 11: Thank you for pointing this out. We agree with your comment and have revised the title of Section 3.4 to ensure it is distinct from Section 3.3. Specifically, we changed the title of Section 3.4 to "3.4 The effect of various hot-air drying temperatures on the flavor profile of Houttuynia cordata Thunb." to better reflect its specific content and to avoid duplication. This change can be found on Line 500 of the revised manuscript and has been highlighted in yellow for your convenience.

Comments 12: I recommend incorporating practical applications derived from the current results for industrial purposes and please highlight the novelty of this research.

Response 12: We sincerely appreciate the reviewer's valuable suggestion to emphasize both the practical industrial applications and scientific novelty of our research. We have strengthened these aspects in the Discussion section to highlight the significance of our findings. The supplementary content can be found on Lines 842-847, and all modifications have been highlighted in yellow.

Comments 13: I would recommend reviewing the formatting of all scientific names throughout the manuscript to ensure they are consistently italicized.

Response 13: We thank the reviewer for this valuable suggestion, which has enhanced the manuscript’s adherence to scientific writing conventions. We examined all Latin names of Houttuynia cordata Thunb. and drying kinetics parameters to ensure that the Latin names were properly italicized.

Comments 14: Please make sure the Introduction and Materials and Methods sections are closely aligned with the Results section. The Introduction should mention the research objectives and rationale, while the Materials and Methods section should provide a detailed description of the experimental procedures, including sample preparation, operating conditions, equipment used, and analytical methods, with appropriate references.

Response 14: Thank you for your insightful comment. We fully agree with your suggestion and have revised both the Introduction and Materials and Methods sections to ensure consistency with the Results section.

In the Materials and Methods section (Lines 124–137), we have expanded the descriptions of the experimental procedures, including detailed sample preparation steps, drying conditions, equipment models, and analytical techniques, along with appropriate references.

Reviewer 2 Report

Comments and Suggestions for Authors

The reviewed manuscript discusses the effect of hot air drying temperature on the chemical composition, microstructure, flavor characteristics and bioactive properties of dried Houttuynia cordata Thunb. The drying kinetics were also analyzed and the process was modeled to assess drying parameters such as effective diffusion coefficient and activation energy.

The topic, idea and research plan are interesting and at the same time cover a very wide range of measurements. This makes it a detailed and valuable supplement to the knowledge in the field of herbaceous plant drying technology. However, in my opinion, the manuscript needs some improvement as noted below.

  1. Given the very broad nature of the study, it is worth considering moving some of the detailed results to supplementary materials (e.g. Table 2). In my opinion, this could increase the coherence and clarity of the manuscript.
  2. All materials and equipment used in the study must be defined in detail (at least the full name with type, manufacturer and country). Many measuring devices have not been specified. Please check this issue and correct it.
  3. The main topic of the study is the hot air drying process, hence its conditions must be precisely defined. The type of dryer, drying air flow direction, drying air flow rate, layer thickness of dried material, etc. are not specified.
  4. How was the end of the drying process determined in each experiment ? What were the final moisture contents in each experiment ?
  5. The main comparative material was freeze-dried HCT, and the freeze dryer and used process conditions were not defined at all. This is important for a complete assessment, especially since no measurements were made for the fresh raw material.
  6. Equation (3) and (4) – “M” and “Mt” have the same values ? If so please unify symbols, if not please define correctly.
  7. Not all methods are sufficiently described in the “Materials and Methods” section. Please try to expand on some issues a little. In addition, some details about the measurement methodology are in the "Results" section, but they should be in the "Material and Methods" section.

Line 53 – “editable” - it should probably be "edible"?

Line 78 – “wind speed” - it would be better to use "air flow rate".

Line 83 – “active ingredients” – it would be better to use “bioactive ingredients”.

Line 217 – “Fllowing” - it should probably be "Following".

Line 330 – “driing” – it should be “drying”.

Author Response

Comments 1: Given the very broad nature of the study, it is worth considering moving some of the detailed results to supplementary materials (e.g. Table 2). In my opinion, this could increase the coherence and clarity of the manuscript.

Response 1: We sincerely appreciate the reviewer’s constructive suggestion to enhance the manuscript’s clarity. We agree that relocating some detailed results to supplementary materials would improve the coherence of the main text. Changes made: Table 2 has been moved to the Supplementary Materials (now labeled Table S1).  A reference to this table has been added in Lines353. 

Comments 2: All materials and equipment used in the study must be defined in detail (at least the full name with type, manufacturer and country). Many measuring devices have not been specified. Please check this issue and correct it.

Response 2: We sincerely appreciate the reviewer's valuable suggestion. We agree that detailed specifications of materials and equipment are essential for reproducibility. Accordingly, we have added the full descriptions of all key instruments used in the study, including drying equipment (Line 125 and 129), UPLC-LTQ-Orbitrap-MS/MS system (Line 216), electronic tongue (Line 266), and full-wavelength spectrophotometer (Line 260-261). We have added full details of all instruments and changes highlighted in yellow. 

Comments 3: The main topic of the study is the hot air drying process, hence its conditions must be precisely defined. The type of dryer, drying air flow direction, drying air flow rate, layer thickness of dried material, etc. are not specified. How was the end of the drying process determined in each experiment? What were the final moisture contents in each experiment?

Response 3: We sincerely appreciate the reviewer's insightful comment regarding the need for more precise description of the hot air drying process. We fully agree that these technical details are essential for reproducibility. Accordingly, we have supplemented the experimental section with comprehensive specifications of the drying conditions, with all modifications highlighted in yellow. (Lines 128-137)

Comments 4: The main comparative material was freeze-dried HCT, and the freeze dryer and used process conditions were not defined at all. This is important for a complete assessment, especially since no measurements were made for the fresh raw material.

Response 4: We greatly appreciate the reviewer's valuable comment regarding the need for complete documentation of the freeze-drying process parameters. We agree that these details are essential for proper comparison between drying methods.

We have supplemented the following detailed freeze-drying specifications in Section 2.2 "Assessment of the drying kinetics of HCT" (Lines 124-127), with all modifications highlighted in yellow. 

Comments 5:

Equation (3) and (4) – “M” and “Mt” have the same values ? If so please unify symbols, if not please define correctly.

Response 5: We sincerely appreciate the reviewer’s attentive feedback. We agree with the observation regarding the symbols M and Mₜ in Equations (3) and (4). These indeed represent the same concept (equilibrium moisture content), and their dual use was unintentional. To avoid confusion, we have unified the notation to consistently use Mₜ throughout the manuscript.Equations (3) and (4) now both use Mₜ (highlighted in yellow).

Updated text: Where Mt is the moisture content at t (g water/g dry matter), t is drying time(min) (Line 142); M(t+dt) is the moisture content at t + dt (g water/g dry matter), respectively, t is drying time(min) (Line 149) 

Comments 6: Not all methods are sufficiently described in the “Materials and Methods” section. Please try to expand on some issues a little. In addition, some details about the measurement methodology are in the "Results" section, but they should be in the "Material and Methods" section.

Response 6: We sincerely appreciate the reviewer's careful reading and constructive suggestions regarding methodological descriptions. We agree that complete method documentation should be consolidated in the "2.2 Materials and Methods" section. All methodological details originally appearing in the Results section regarding drying kinetics measurement procedures have been moved to Section 2.2 (Lines 128-137), highlighted in yellow.

 Comments 7: Line 53 – “editable” - it should probably be "edible"?

Response 7: We sincerely appreciate the reviewer's meticulous attention to detail in identifying this terminology issue. We agree with the suggested correction and have implemented the change accordingly in Line 57 of the revised manuscript. The revision has been highlighted in yellow for easy reference. 

Comments 8: Line 78 – “wind speed” - it would be better to use "air flow rate".

Response 8: We appreciate the reviewer's precise suggestion regarding terminology standardization. We agree that "air flow rate" is the more scientifically appropriate term in this context. The term "wind speed" has been replaced with "air flow rate" throughout the manuscript (Line 84, highlighted in yellow) 

Comments 9: Line 83 – “active ingredients” – it would be better to use “bioactive ingredients”.

Response 9: We sincerely appreciate the reviewer's precise suggestion regarding terminology improvement. We agree that "bioactive ingredients" is a more scientifically accurate term in this context. The term "active ingredients" has been replaced with "bioactive ingredients" throughout the manuscript where appropriate specifically in Line 90. 

Comments 10: Line 217 – “Fllowing” - it should probably be "Following".

Response 10: We sincerely appreciate the reviewer’s careful attention to detail in identifying this typographical error. The word “Fllowing” has been corrected to “Following” in Line 253.The correction has been highlighted in yellow for easy reference. 

Comments 11: Line 330 – “driing” – it should be “drying”.

Response 11: We sincerely appreciate the reviewer's meticulous attention to detail in identifying this typographical error. We have carefully corrected this mistake and verified the entire manuscript for similar issues. Corrected "driing" to "drying" in Line 365. The modification has been highlighted in yellow for easy identification

Reviewer 3 Report

Comments and Suggestions for Authors

Abstract section: Please consider adding the temperatures that were evaluated.

Methodology section: The authors must include a subsection describing the drying process, including details such as the temperatures used, the equipment employed, and other relevant parameters.

Section 2.1: Please specify the geometry of the samples. This information is critical for the mathematical modeling of thin layer drying. Additionally, provide the value of L.

Lines 153 to 156: Please specify the parameter m and note that L0 is not included in the equations.

Figure 1: Please review the y-axis labels in all four figures.

Lines 322 to 325: What were the Deff values at the different temperatures?

Lines 558 and 562. Check subscripts.

Discussion section:

Lines 640 to 646: Please enhance the discussion regarding moisture diffusion. Consider adding comparative information from other vegetal matrices to enrich the analysis.

Line 650: The statement “ low Ea” is relative, please clarify what is considered a low value of activation energy in this context.

Refernce section: More than 40% of the references are over 10 years old. The authors should consider updating the reference list by including more recent and relevant literature.
